# Optoelectronic system and device integration for quantum-dot light-emitting diode white lighting with computational design framework

Chatura Samarakoon [1,2], Hyung Woo Choi[1,2], Sanghyo Lee[1,2], Xiang-Bing Fan[1], Dong-Wook Shin[1], Sang Yun Bang [1], Jeong-Wan Jo[1], Limeng Ni[1], Jiajie Yang[1], Yoonwoo Kim[1], Sung-Min Jung [1 ✉], Luigi G. Occhipinti [1], Gehan A. J. Amaratunga[1] & Jong Min Kim [1]

We propose a computational design framework to design the architecture of a white lighting system having multiple pixelated patterns of electric-field-driven quantum dot light-emitting diodes. The quantum dot of the white lighting system has been optimised by a system-level combinatorial colour optimisation process with the Nelder-Mead algorithm used for machine learning. The layout of quantum dot patterns is designed precisely using rigorous device-level charge transport simulation with an electric-field dependent charge injection model. A theoretical maximum of 97% colour rendering index has been achieved with red, green, cyan, and blue quantum dot light-emitting diodes as primary colours. The white lighting system has been fabricated using the transfer printing technique to validate the computational design framework. It exhibits excellent lighting performance of 92% colour rendering index and wide colour temperature variation from 1612 K to 8903 K with only the four pixelated quantum dots as primary.

[1] Electrical Engineering Division, Department of Engineering, University of Cambridge, 9 JJ Thomson Avenue, Cambridge CB3 0FA, United Kingdom. [2] These authors contributed equally: Chatura Samarakoon, Hyung Woo Choi, Sanghyo Lee. ✉email: sj569@cam.ac.uk

D riven by technology advances in solid-state optoelec-
tronics, light-emitting diode (LED) lighting has been
expanded to a wide range of applications in consumer
electronics, automotive, architectural, and healthcare settings[1–4].
Recently, there has been a drive to make lighting "smart" by
incorporating internet of things (IoT) connectivity and machine
learning-based artificial intelligence (AI)[5,6]. Next-generation
smart lighting is expected to be able to respond to personal
mood or circadian rhythm in the daily ambient living environ-
ment while also rendering the colours of objects more
accurately[7–10]. In order to achieve these requirements, it is
essential for smart lighting to have high colour rendering cap-
ability and wide colour controllability[11,12].

Electroluminescence (EL)-based quantum dot light-emitting
diodes (QD-LEDs) are suitable devices for next-generation smart
lighting[13–17]. QD-LEDs exhibit high colour purity due to the
narrow bandwidths of their optical emissions. The emission peak
wavelengths can also be controlled readily by tuning their optical
bandgaps[18,19]. This makes QD-LEDs an excellent platform for
precisely controlling the electro-optical properties of white
lighting systems. A white lighting system is characterised by the
colour rendering index (CRI) and the correlated colour tem-
perature (CCT)[20]. The emission spectrum of the white lighting
system can be controlled by employing multiple pixelated pat-
terns of colour tunable QDs in a single lighting device. By opti-
mising the emission spectrum of a white lighting system in
conjunction with the appropriate pixel layout of the emission
patterns of QD-LEDs, a high CRI and wide colour controllability
can be achieved in principle.

One of the practical system architectures that enable the
integration of multiple pixelated QD-LED devices into a single
lighting system is based upon patterned QD-LEDs[21–23]. The
multiple pixelated patterns of QDs can be realised by a transfer
printing technique to define different light emission areas on the
device substrate[24–26]. However, the problem of optimising the
light emission architecture of a white lighting system remains
unresolved, as it requires an excessive amount of colour calcu-
lations for the massive combinations of QD colours and their
geometrical combinations available for pixel layout. To reduce the
number of colour evaluations for the optimal layout design, one
of the systematic and efficient approaches is to employ compu-
tational design by way of combinatorial colour optimisation and
accurate charge transport simulation of QD-LED devices[27–30].

In this study, we propose a computational design framework to
optimise a white lighting system configured by $N$-number of
pixelated patterns of QD-LED devices. The computational design
framework is divided into two parts: (i) a system-level design of
lighting system by a combinatorial colour optimisation process
with a numerical optimisation algorithm used for machine
learning, and (ii) a device-level layout design of multiple pixelated
QD-LEDs using a charge transport simulation based on an
electric-field dependent charge injection model, with the device
and material-level parameters obtained from experiment.

First, system-level lighting colour properties are optimised by a
colour optimisation process based on a combinatorial approach
to finding the optimal QD combination for the emission spec-
trum satisfying a high CRI with a chromaticity close to the
Commission Internationale de l'éclairage (CIE) standard illumi-
nant D65. In the colour optimisation process, the Nelder-Mead
algorithm commonly used for optimising a neural network of
machine learning is utilised as a combinatorial method to reduce
the number of colour evaluations for the massive possible com-
bination of QDs[31–34]. Here, a cost function, that is a function of
the CRI and chromaticity for a given emission spectrum, is
defined as an assessment metric of lighting properties and
minimised by the Nelder-Mead algorithm to search the optimal

QD combination for the emission spectrum satisfying a high CRI
and the chromaticity of CIE D65 illuminant, simultaneously.

Next, the device-level simulation of QD-LEDs is carried out by
a complete charge transport model with a detailed electric-field
dependant carrier injection model[27,29,30,35] to predict the electro-
optical properties including the intrinsic spectral radiances of the
monochromatic QD-LED devices. The optoelectronic simulations
are extended to design the layout of the multiple pixelated QD-
LEDs. Here, material-level parameters are experimentally
extracted from fabricated QD-LED devices and transmission
electron microscopy (TEM) analysis of the QD particles. From
the device-level simulation of our charge transport model, the
emission pattern layouts of the pixelated QD-LED device are
designed to fulfil the system-level emission spectrum of the
lighting system. Here, the layout design rule determining the
emission widths of pixelated QD-LEDs is developed from
underlying physical parameters. As a result of our computational
design, the theoretical lighting properties of 97% CRI with a wide
colour variation of CCT ranging from 2243 K to 9207 K are
achieved by only four primary colours of red, green, cyan, and
blue QDs with an optimal layout of pixelated emission patterns.

Finally, to validate our computational design framework, we
fabricated a white lighting system with the 4-primary colours of
red, green, cyan, and blue pixelated QDs using a transfer printing
technique. Here, cadmium selenide (CdSe) based QDs are used
with precise control of emission peak wavelengths. The experi-
mental results agree very well with the simulation results, exhi-
biting excellent lighting performance of 92% CRI with wide
colour controllability in CCT variation ranging from 1612 K to
8903 K with four pixelated EL-driven QD-LEDs integrated into
the white lighting system.

## Results
**Lighting system architecture and computational design fra-
mework.** In order to achieve the lighting properties suitable for
smart white lightings, the system-level colour optimisation of
lighting system and the device-level design of individual QD-LED
are carried out. First, a light emission spectrum that satisfies high
CRI with colour chromaticity close to CIE D65 illuminant should
be identified through the system-level colour optimisation pro-
cess. Figure 1a shows the architecture of a lighting system with $N$
number of stripe-patterned QD-LED pixels. The lighting system
is configured as a stack structure having a hole injection layer
(HIL), a hole transport layer (HTL), an emissive layer (EML) and
an electron transport layer (ETL) sandwiched between anode and
cathode electrodes. The $N$ number of the different pixelated QD
stripe-patterns are formed between the HTL and the ETL as the
EML. Each of the pixelated QD patterns has a different emission
peak wavelength $\lambda$ and pattern width $w$ to determine its emission
peak radiance. To maximise the CRI with a targeted CIE D65
chromaticity, a set of peak wavelengths $\Lambda = (\lambda_1, \lambda_2, …, \lambda_N)$ and a
set of emission widths $\mathbf{w} = (w_1, w_2, …, w_N)$ should be optimised
for the layout of the pixelated QD-LEDs.

The first design step is to determine the set of peak wavelengths
$\Lambda$ for the multiple pixelated QD-LEDs using a computational
colour optimisation process (Fig. 1b–e). Figure 1b shows an
example emission spectrum $L(\lambda, \Lambda)$ of a lighting system with the
set of peak wavelengths $\Lambda$ and the set of normalised emission
powers $\boldsymbol{\beta} = (\beta_1, \beta_2, …, \beta_N)$ (see the Methods section). Each of the
normalised power $\beta_n$ for a peak wavelength $\lambda_n$ of $n$-th QD, where
$n$ is a natural number from 1 to $N$, is designed to be $\beta_n = D65(\lambda_n)$
which is the normalised power distribution of the CIE D65
illuminant used as a reference illuminant. Then, the CRIs for the
emission spectrum are calculated over the reflectance spectra of
14 test colour samples (TCSs) (Fig. 1c, d). The chromaticity

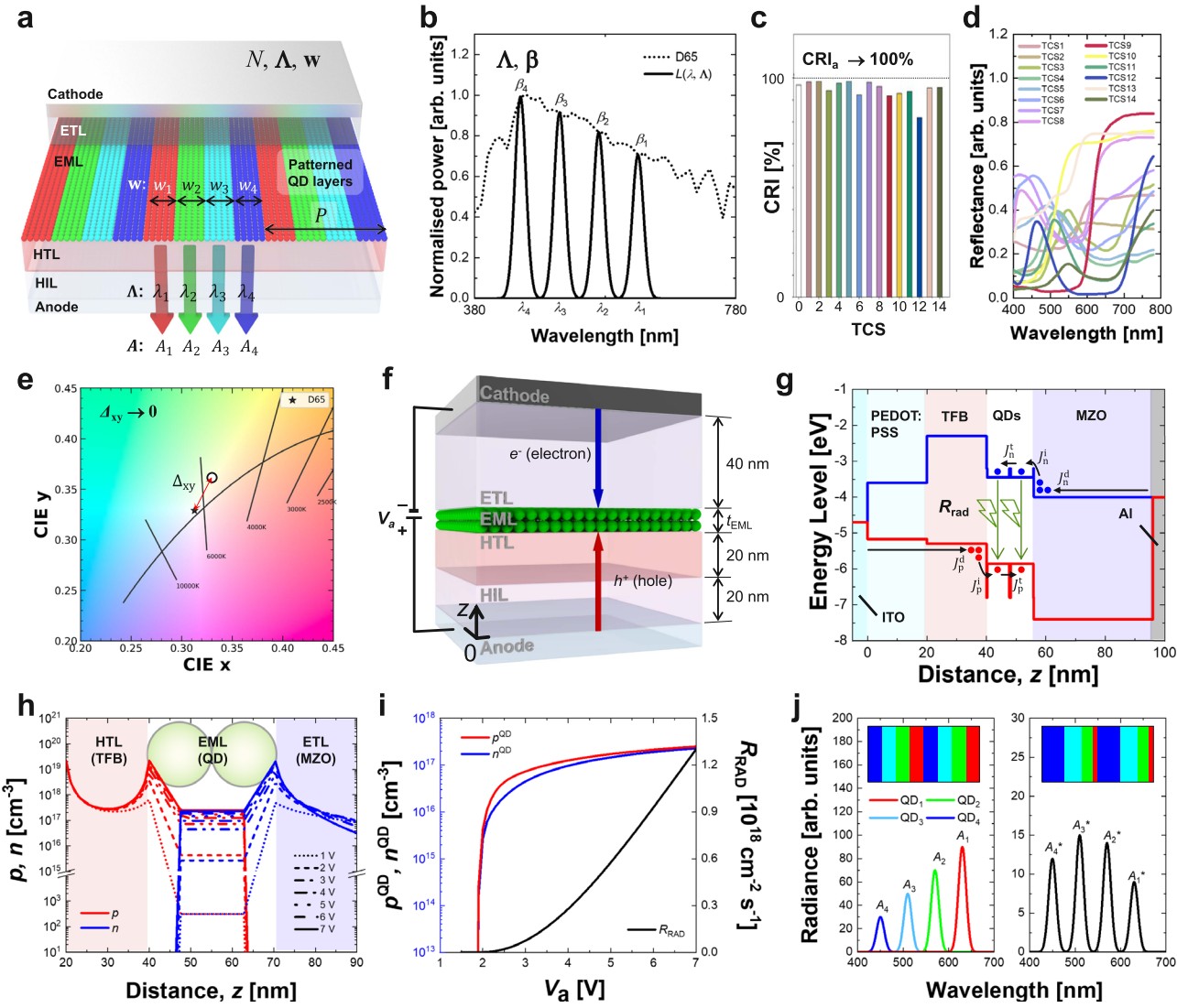

**Fig. 1 A system architecture and design procedure of multi-primary coloured lighting system with patterned QD-LEDs. a** $N$-primary coloured lighting system. $N$ is the number of primary colours; $\Lambda = (\lambda_1, \lambda_2, ..., \lambda_N)$ is a set of peak wavelengths ($\lambda_i$), $\mathbf{w} = (w_1, w_2, ..., w_N)$ is a set of emission widths ($w_i$), and $\mathbf{A} = (A_1, A_2, ..., A_N)$ is a set of peak radiances ($A_i$) for $i$-th QD pattern. $P$ is the pitch of the pixel group. ETL electron transport layer, EML emissive layer, HTL hole transport layer, HIL hole injection layer. **b** An emission spectrum of white lighting system to be optimised by the colour optimisation process. D65 is a normalised power distribution of the CIE D65 standard illuminant, and $L$ is a power distribution of any given white lighting system. $\boldsymbol{\beta} = (\beta_1, \beta_2, ..., \beta_N)$ is a set of peak powers ($\beta_i$) normalised for the D65 power distribution. **c** Exampled CRI values of a given emission spectrum over the test colour samples (TCSs). $CRI_a$ is the average CRI of TCSs from 1 to 8. **d** Reflectance spectra of 14 TCSs. **e** Chromaticity difference $\Delta_{xy}$ from the CIE D65 illuminant. **f** A stratified layer structure of a monochromatic QD-LED device used in the charge transport simulation. $z$ spatial variable, $t_{EML}$ thickness of EML. **g** A flat-band energy-level diagram across the device with respect to the distance from the anode electrode. ITO indium tin oxide, TFB Poly[(9,9-dioctylfluorenyl-2,7-diyl)-co-(4,4'-(N-(4-sec-butylphenyl)diphenylamine)], QDs quantum dots, MZO magnesium-doped zinc oxide, Al aluminium. PEDOT:PSS is a poly(3,4-ethylenedioxythiophene)-poly(styrenesulfonate). $J_p^d$, $J_n^d$, $J_p^i$, $J_n^i$, $J_p^t$, and $J_n^t$ are drift-diffusion, injection, and tunnelling current densities for holes (subscript p) and electrons (subscript n). $R_{RAD}$ is a radiative recombination rate per unit area. **h** Hole and electron density distributions for the applied voltages. **i** Simulated voltage dependencies of hole density ($p^{QD}$), electron density ($n^{QD}$), and $R_{RAD}$ at QD layer. **j** Schematic illustration of the emission peak adjustment by changing pattern widths. $A_i^*$ is adjusted peak radiance of $i$-th QD pattern. Insets are the emission pattern layouts of the devices with identical emission widths and different emission widths.

difference $\Delta_{xy}$ of the emission spectrum is also calculated as the distance from the chromaticity coordinates of the CIE D65 illuminant as depicted in Fig. 1e. In order to optimise the combination of QDs, we developed a cost function that is a function of an average CRI and a chromaticity difference of an emission spectrum for a given QD combination. The cost function is effectively minimised by the Nelder-Mead algorithm as a combinatorial approach which enables fewer computational colour evaluations, ensuring that the emission spectrum for the optimised QD combination can have an average CRI close to

100% and $\Delta_{xy}$ close to zero. Here, the optimal set of peak wavelengths $\mathbf{w}$ and its corresponding set of normalised emission powers $\boldsymbol{\beta}$ are simultaneously determined by our colour optimisation process.

Once the system-level optimal emission spectrum of lighting system is obtained, the next step is a device-level design to determine the emission widths of pixelated QD patterns integrated into the white lighting system. For the optimal emission spectrum, there are two ways to control the emission contribution of each QD patterns. One is to optimise the QD

materials for identical emission widths of QD patterns, and the other is to optimise the emission widths of QD patterns for the given QD materials. Since it is difficult to control the material parameters of QDs for the identical emission widths, the emission widths of the pixelated QD patterns are optimised for a given QD materials. For the device-level design, the electro-optical properties including the emission spectrum of the monochromatic device are predicted by a charge transport simulation as depicted in Fig. 1f, g. The stack structure of the QD-LED device used in this study is illustrated in Fig. 1f. The thicknesses of transport layers are taken to be 40 nm, 20 nm, and 20 nm for the ETL, HTL, and HIL, respectively, and the thickness of the EML ($t_{EML}$) is determined by the diameters of QDs and the number of QD layers. A flat-band energy-level diagram with respect to the distance from the anode electrode is shown in Fig. 1g. An indium tin oxide (ITO), a Poly(3,4-ethylenedioxythiophene)-poly(styrenesulfonate) (PEDOT:PSS), a Poly[(9,9-dioctylfluorenyl-2,7-diyl)-co-(4,4′-(N-(4-sec-butylphenyl)diphenylamine)] (TFB), QDs, a magnesium-doped zinc oxide (MZO), and an aluminium (Al) are used for the anode electrode, HIL, HTL, EML, ETL, and cathode electrode, respectively. CdSe and zinc sulphide (ZnS) are used as core and shell materials of the QDs, respectively. The material parameters for each of the transport layers and QD shell are listed in Supplementary Table 1.

The schematic pathways for possible current flow across the QD-LED devices are also depicted in Fig. 1g. The charge transports across the device can be described by (i) the drift-diffusion current densities $J_p{}^d$ and $J_n{}^d$ in the HIL/HTL and ETLs, (ii) the tunnelling current densities $J_p{}^t$ and $J_n{}^t$ between two neighbouring QD layers, and (iii) the injection current densities $J_p{}^i$ and $J_n{}^i$ between the QD layer and charge transport layers[35]. Especially for the charge injection, the holes and electrons are injected to the QD layer from the transport layer by the electric-field-dependent carrier capturing process[30]. The holes and electrons injected into the QD layers form electron-hole pairs, and photons in the visible range for the optical bandgap of the QDs are generated in the EML by the radiative recombination process of the electron-hole pairs with a radiative recombination rate per unit area, $R_{RAD}$[27,30,36]. The simulated hole and electron densities, $p$ and $n$, across the device under different voltages are plotted in Fig. 1h as an example of the simulation result. The detailed charge transport model is described in the Method section.

As an example of the simulation, the hole and electron densities at the QD layer, $p^{QD}$ and $n^{QD}$ and the resulting radiative recombination rate per unit area, $R_{RAD}$ with respect to the applied voltage are shown in Fig. 1i. Using the predicted radiative recombination rate per unit area, $R_{RAD}$, the layout of the emission patterns of QD-LEDs is designed as illustrated in Fig. 1j. First, the spectral radiances of individual monochromatic QD-LED at a given voltage are calculated from $R_{RAD}$ of each device. Here, different QD-LEDs show different emission spectra due to their different physical properties, even if the emission areas are the same as shown in the left inset in Fig. 1j. The overall emission spectrum of the white lighting system configured with the patterned QD-LEDs is obtained by summing up the spectral radiances of individual QD-LED with weighting factors determining each emission width of QD-LED pattern. Therefore, by designing the weighting factors of each emission pattern based on the charge transport simulation, the emission spectrum of the white lighting system can be matched to the emission spectrum optimised by the aforementioned colour optimisation process, as shown in the right inset in Fig. 1j.

Our computational framework can also be applied to the various types of system architectures such as patterned[21,24], stacked[22,23], and mixed[15,37] types of QD-LEDs fabricated by different process techniques for various lighting systems. The

spectrum of any kind of the QD-LED white lighting system with multiple primary colours can be systematically designed by our combinatorial colour optimisation process. The charge transport simulation can also be applied to the different types of lighting system architecture with the simulation models modified for their QD configurations. The computational design framework can be further used for the system design of the EL-based full-colour QD-LED displays with multiple primary colours. The colour optimisation process can be used for optimising the white balance, and the charge transport model can be used to predict the EL characteristics of each QD-LED subpixel with respect to the applied voltage. In summary, the computational design framework can be used for various smart lighting and display applications with similar operational principle.

**System-level colour optimisation process with Nelder-Mead method.** To fulfil the requirements of the smart white lightings, a system-level design based on the combinatorial colour optimisation process are performed in this section. Figure 2a shows the detailed flow chart of the iterative procedure used to search the optimal set of peak wavelengths $\Lambda^{opt}$ for a lighting system with a QD combination of $N$ primary colours. First, the cost function $f$ encoding the properties of CRI and the CIE chromaticity is defined by Eq. (1).

$$f(\Lambda) = 1.5\|\mathbf{CIE}_{xy}(L(\lambda, \Lambda)) - \mathbf{CIE}_{xy}(D65)\|_2^2 - \left(\frac{CRI_a(L(\lambda, \Lambda))}{100} - 1\right)$$

(1)

Here, $\mathbf{CIE}_{xy}$ is the CIE chromaticity coordinates and $CRI_a$ is the average CRI over 8 TCSs of the spectrum $L(\lambda, \Lambda)$ for any given QD combination[20]. As defined in Eq. (1), the smaller the cost function is, the higher the average CRI and the smaller the chromaticity difference become. If the cost function evaluation is performed via a simple grid searching process with 1 nm step in the range from 400 nm to 700 nm, it takes ~$300^N$ cost function evaluations ($7.29 \times 10^{14}$ times of evaluations for $N = 6$), which is prohibitively computationally expensive. Therefore, in order to efficiently optimise the QD combination with reduced amount of calculation, the Nelder-Mead algorithm is used as a combinatorial approach to minimise the cost function by tunning the set of peak wavelengths $\Lambda$. By the Nelder-Mead minimisation, the optimal set of peak wavelengths $\Lambda$ and its corresponding set of normalised emission powers $\beta$ for the given $N$ number of primary colours are simultaneously determined.

The Nelder-Mead method is used as the iterative optimiser in the combinatorial colour optimisation process to search an optimal peak wavelength set that minimises the cost function for the optimality conditions in a $N$-dimensional peak wavelength space[31–33]. The Nelder-Mead algorithm in the iterative procedure is illustrated in Fig. 2b for $N = 2$ as an example. The Nelder-Mead algorithm uses the group of $N + 1$ peak wavelength sets for $N$ primary colour system. At the $k$-th step of iteration process, a group of the peak wavelength sets $\Lambda_m{}^k$ for $m = 1, 2, ..., N + 1$ are used to calculate the group of the cost function values $f_m{}^k$ according to Eq. (1). From the group of wavelength sets $\Lambda_m{}^k$, a new group of wavelength sets $\Lambda_m{}^{k+1}$ for the $(k + 1)$-th step is obtained by the reflection, expansion, shrinkage, and contraction operations in the Nelder-Mead algorithm. This process is repeated till the cost function reaches a certain convergence level $C_{min}$, and the optimised wavelength set $\Lambda^{opt}$ is finally determined by the end of the iterative procedure. Figure 2c shows the convergence results of the cost functions during the iterative optimisation procedure for 3-, 4-, 5-, and 6-primary colours. The cost functions converge to their own minimum levels, and the Nelder-Mead method successfully finds an optimal set of

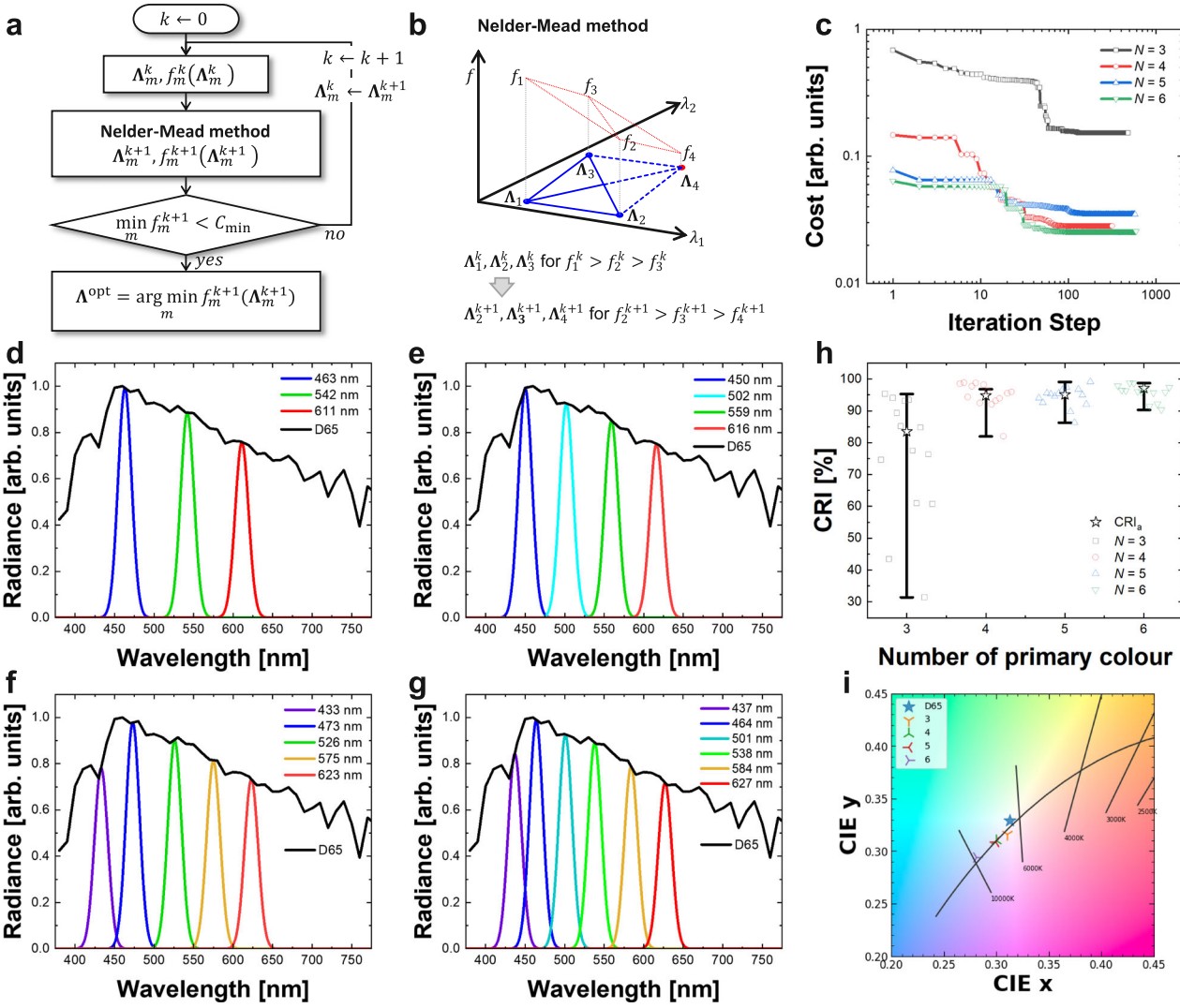

**Fig. 2 Optimal emission spectra of multi-primary lightings designed by the colour optimisation process. a** Iterative optimisation procedure determining a peak wavelength set of $N$-primary lighting. An index for a given iteration step is denoted by $k$. $\Lambda_m^k$ is a $m$-th peak wavelength set in the group of the peak wavelength sets for the Neler-Mead method, where $m \in \{1, 2, ..., N+1\}$. $f_m^k(\Lambda_m^k)$ is a cost for the given peak wavelength set. 'min $f_m^{k'}$ is a minimum value among the costs $f_m^k$, and $C_{min}$ is a criteria of convergence level for stopping the iteration process. 'arg min' is an argument of the minimum for the cost function. **b** Schematic illustration of the Nelder-Mead algorithm minimising the cost function. **c** Convergence plots of the cost functions for $N$-primary lightings. The cost functions are minimised to have high CRI and chromaticity close to D65 illuminant. **d–g** Spectra of 3-, 4-, 5-, and 6-primary lightings for their optimised peak wavelength sets. The emission peaks at each peak wavelength are designed to follow the normalised power distribution of the CIE D65 illuminant. **h** CRI values of $N$-primary lightings with the optimised peak wavelength sets. Individual CRIs for 14 TCSs are marked with rectangles, circles, triangles, and inverse triangles for the optimised 3-, 4-, 5-, and 6-primary lightings. The average CRI (CRI$_a$) for TCSs from 1 to 8 and the range of the CRIs are marked with star and minimum/maximum interval plot. **i** CIE chromaticity coordinates for the 3-, 4-, 5-, and 6-primary lightings designed by the colour optimisation process.

wavelengths for each multi-primary white lighting system. The method takes a few hundreds of cost function evaluations through the iteration steps. Thus, the optimal wavelength set is efficiently obtained with a practical computational speed.

The peak wavelength sets of 3-, 4-, 5-, and 6-primary colour lighting systems have been optimised by the colour optimisation process, and their resultant CRIs and chromaticity are comprehensively analysed in Fig. 2d–i. Figure 2d–g show the emission spectra for 3-, 4-, 5- and 6-primary lighting systems optimised by the colour optimisation process. Throughout the colour optimisation process for each of the 3-, 4-, 5-, and 6-primary coloured lighting, the primary colours and their peak wavelengths were optimised as follows. For the 3-primary lighting, red (611 nm), green (542 nm), and blue (463 nm) are selected as primary

colours (Fig. 2d), while the 4-primary lighting is optimised to have the primary colours of red (616 nm), green (559 nm), cyan (502 nm), and blue (450 nm) (Fig. 2e). In the case of the 5-primary lighting, the primary colours are chosen to be red (623 nm), yellow (575 nm), green (526 nm), blue (473 nm), and purple (433 nm) (Fig. 2f). Finally, the 6-primary lighting is optimised with the primary colours of red (627 nm), yellow (584 nm), green (538 nm), cyan (501 nm), blue (464 nm), and purple (437 nm) (Fig. 2g). In the figures, the peak radiances of the spectra, corresponding to their peak wavelengths, were determined to match the power distribution of the CIE D65 illuminant at a given peak wavelength.

The average CRI and the range of CRI values for each of the optimised $N$-primary lighting system are summarised in Fig. 2h.

**Table 1 Chromatic properties of the multi-primary white lightings optimised in this study.**

| Illuminant | Number of primary | CRI [%] | | | CIE$_{xy}$ | $\Delta_{xy}$ |
|---|---|---|---|---|---|---|
| | | average | maximum | minimum | | |
| D65 | - | 100 | 100 | 100 | (0.313, 0.329) | 0.000 |
| Multi-primary QD-LED white lighting | 3 | 84 | 95 | 31 | (0.311, 0.319) | 0.010 |
| | 4 | 97 | 99 | 82 | (0.299, 0.312) | 0.022 |
| | 5 | 95 | 99 | 86 | (0.303, 0.310) | 0.021 |
| | 6 | 97 | 99 | 90 | (0.281, 0.299) | 0.044 |

**CIE$_{xy}$** is the chromaticity coordinates in the CIE colour space and $\Delta_{xy}$ is the chromaticity difference from the CIE standard illuminant D65.

The CRI values of 3-, 4-, 5-, and 6-primary lightings for the 14 individual TCSs are plotted in Supplementary Fig. 1. For the 3-, 4-, 5- and 6-primary lights, the average CRI for the first 8 TCSs were observed to be 84%, 97%, 95% and 97%, respectively. Here, the 4-, 5-, 6-primary lights have a minimum CRI of at least 80% for all TCSs, while the 3-primary light shows a minimum CRI value of 31%. Therefore, a white lighting system with more than four primary colours should be used to have an average CRI of over 95% with minimum CRI over 80%. The chromaticity of each optimised N-primary white lighting system is depicted in Fig. 2i. For the 3-, 4-, 5-, 6-primaries, the chromaticity differences were calculated to be 0.010, 0.022, 0.021 and 0.044, respectively. The lighting systems having 3-, 4-, and 5-primary colours show a chromaticity close to D65, while the 6-primary lighting system shows a substantially larger deviation from D65. The CRI and chromaticity of all the optimised N-primary white lighting systems are summarised in Table 1. As the number of primary colours increases, the lighting performance improves but more numbers of primary QDs are required. Therefore, considering maximal lighting performance with minimal number of primary QDs, four primary colours of red, green, cyan, and blue are sufficient to achieve a high CRI and a chromaticity close to the CIE D65 illuminant. The 4-primary colour configuration chosen by the combinatorial colour optimisation process exhibits an average CRI of 97% and a chromaticity difference of 0.022, with the peak wavelength set of (616, 559, 502, 450) nm and the peak radiance set of (0.75, 0.85, 0.93, 0.99) for red, green, cyan, and blue QD emission patterns, respectively.

**Device-level simulation with charge transport model.** To predict precisely the electro-optical characteristics of QD-LEDs by the device-level charge transport simulation, the material-level parameters should be extracted in advance by fabricating monochromatic QD-LED devices via transfer printing technique with the red, green, cyan, and blue QDs selected in the previous section. Figure 3a–c show the experimental luminance, current density, and external quantum efficiency (EQE) of the fabricated devices with respect to voltage and current density. In the fabrication, we used a CdSe-based QD materials with ZnS shell for red, green, cyan, and blue QDs. The peak wavelengths of the photoluminescence (PL) spectrum of the red, green, cyan, and blue QDs are measured to be 621 nm, 558 nm, 502 nm, and 452 nm, respectively, which are very close to our designed peak wavelengths (Supplementary Fig. 2). In Fig. 3a, the red, green, cyan, and blue QD-LED devices show the luminances of 19,670 cd m$^{-2}$, 10,370 cd m$^{-2}$, 8070 cd m$^{-2}$, and 442 cd m$^{-2}$ at 7 V, respectively. The threshold voltages of the red, green, cyan, and blue QD-LED devices were observed to be 2.0 V, 2.2 V, 2.6 V, and 3.0 V, respectively, and the maximum EQEs of the red, green, cyan, and blue QD-LEDs were also measured to be 4.3%, 0.6%, 1.0%, and 0.3% at 3.2 V, 4 V, 6 V, and 5. 4 V, respectively (Fig. 3b). The current densities at the maximum EQE are

measured to be 0.026 A cm$^{-2}$, 0.10 A cm$^{-2}$, 0.176 A cm$^{-2}$, and 0.158 A cm$^{-2}$, for the respective devices (Fig. 3c).

In terms of device EQE, even though the transfer-printed QD-LED devices show lower EQE than those of the spin-coated devices, they are comparable to the state-of-the-art transfer-printed devices (Supplementary Table 2). The low EQEs of the individual devices can affect the performance of the white lighting system. Therefore, it is required to enhance the EQE of the transfer-printed QD-LED devices by material-level optimisation of QD nanoparticles and device-level customisation of transport layers[22,38,39]. Furthermore, by the process-level innovation of the transfer printing technique and the system-level independent pixel driving method, the white lighting system with enhanced EL performance can be expected[40–42].

The fabricated red, green, cyan, and blue QD-LEDs show different maximum EQE due to differences in their material-level and device-level parameters affected by the QD materials and device architectures. Specifically, the reason why the different EQEs are observed in the individual colours of the QD-LED device is that the QD layers have different properties in terms of both the photoluminescence quantum yield (PLQY) $\eta_{QD}$ and the quality factor Q (Methods and Supplementary Table 3). The PLQY of QD layer as a material- and device-level parameter is affected by inter-particle fluorescence resonance energy transfer (FRET) within the QD layer or at the interfaces of the QD layer for the surrounding transport layers[26,43]. The quality factor as a material-level parameter is determined by the recombination constants which are significantly affected by the defective trap density and the photon interactions in the QD nanoparticle[44]. Therefore, the differences in device EQE are attributed to the different material parameters and interfacial properties of QDs.

Regarding the stability of the transfer-printed device, our fabricated red QD-LED shows a lifetime $T_{50}$@100 cd m$^{-2}$ (time to reach 50% of an initial luminance of 100 cd m$^{-2}$) of 40,000 hours (Supplementary Fig. 3). This is the best stability performance reported to date for transfer-printed QD-LED devices (Supplementary Table 4). Nevertheless, further improvements on the lifetime of our transfer-printed QD-LED device are still needed to lift the stability towards the level of spin-coated QD-LED devices. There are possible pathways to improve the stability of the QD-LED device. One is to optimise the core, shell and ligand materials of QD nanoparticles during synthesis to reduce the surface traps and to enhance the charge confinement effect[26]. Another is to isolate the QD layer from air contact during the transfer printing process since the oxygen or moisture in the air induces surface traps leading to device degradation[45,46].

Figure 3d shows the particle sizes of the CdSe/ZnS QDs measured using TEM analysis. The average diameters of the red, green, cyan, and blue QDs used for the device fabrication were 15.2 nm, 8.4 nm, 7.5 nm, and 6.9 nm, respectively (Fig. 3d and Supplementary Fig. 4). Insets are the snapshots of the fabricated EL-driven monochromatic red, green, cyan, and blue QD-LED devices. The monochromatic devices are fabricated by the transfer

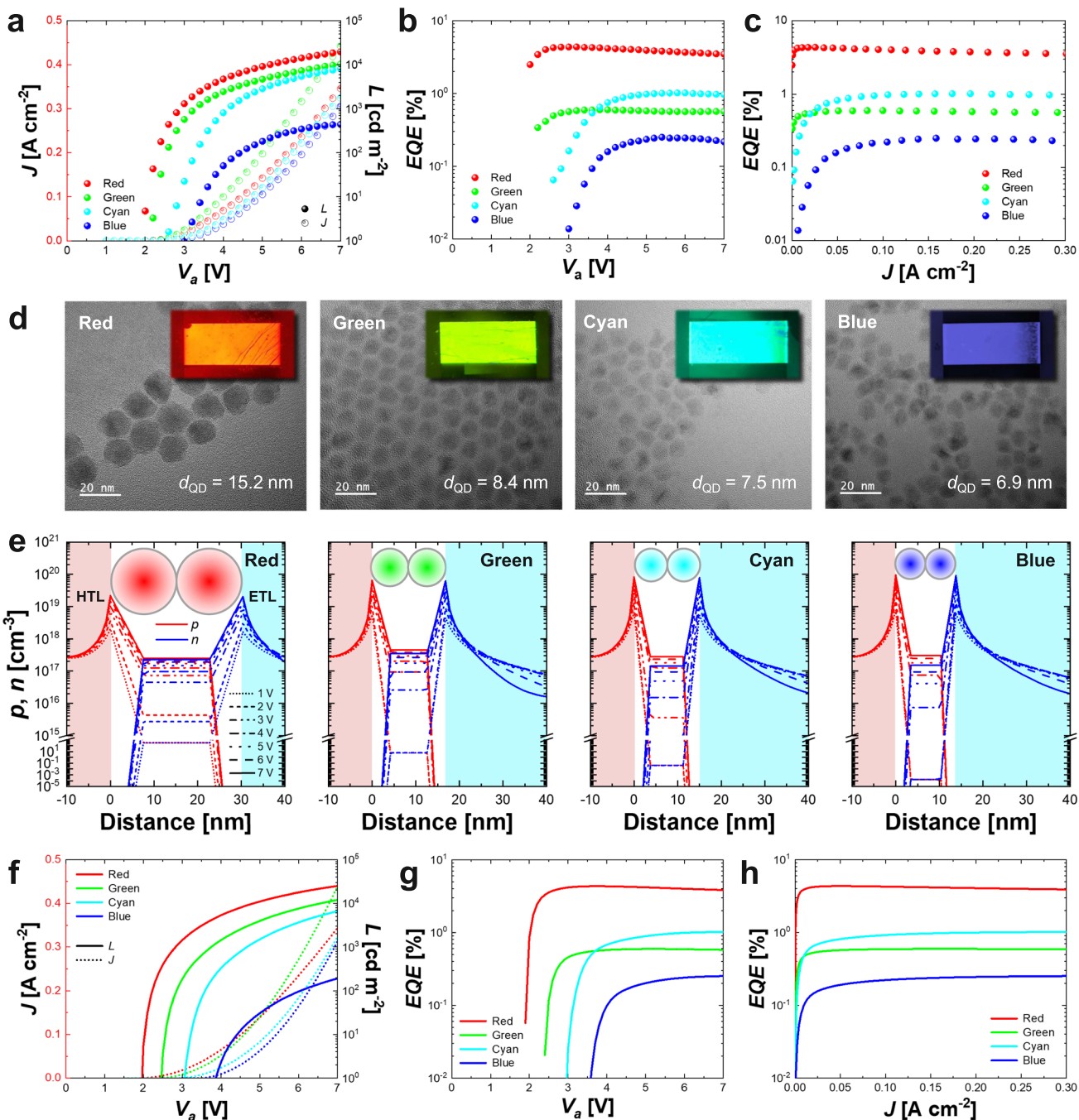

**Fig. 3 Electro-optical characteristics of monochromatic QD-LEDs obtained from the experiments and the charge transport simulation. a** Voltage ($V_a$) dependencies of current density ($J$) and luminance ($L$) of fabricated red, green, cyan, and blue monochromatic QD-LED devices. **b, c** EQE curves of the fabricated monochromatic devices for the voltage (**b**) and the current density (**c**). **d** TEM images for the particle sizes of red, green, cyan, and blue QDs used for the device fabrication and charge transport simulation. $d_{QD}$ is the average diameter of the QD nanoparticles. Insets are the snapshots of EL-driven monochromatic red, green, cyan, and blue QD-LED devices fabricated by the transfer printing technique. The size of the fabricated device is $3.0 \times 1.5$ mm². **e** Simulated hole and electron density distributions around the QD layers for red, green, cyan, and blue QD-LED devices for the applied voltages from 1 V to 7 V with 1 V step. **f** Simulated voltage dependencies of current density and luminance of QD-LED devices. **g, h** Simulated EQE curves for the voltage (**g**) and the current density (**h**). The simulation parameters are extracted with the charge transport model by emulating the experimental EQE-current density curves of the fabricated monochromatic QD-LEDs.

printing technique with an emission size of $3.0 \times 1.5$ mm². For the particle sizes and the EQE-current density curves of each QD-LEDs given in Fig. 3c, the material-level parameters such as the Shockley-Read-Hall (SRH) recombination lifetime $\tau$ and the Auger capture probability $C$ of individual QD-LED device were obtained by matching the simulated EQE-current density curves with the experimental EQE-current density curves in Fig. 3c.

These QD parameters extracted from the experiments and simulation are listed in Supplementary Table 3.

Figure 3e shows the simulated hole and electron density distributions near the QD layers of the red, green, cyan, and blue QD-LED devices for different voltages. The theoretical energy bandgaps of the red (616 nm), green (559 nm), cyan (502 nm), and blue (450 nm) QDs used in the simulation are calculated to

be 1.98 eV, 2.20 eV, 2.44 eV, and 2.72 eV, for their respective peak wavelengths. For the simulation, the EML thicknesses are set to be 30.4 nm, 16.8 nm, 15.0 nm, and 13.8 nm for the respective QD diameters of red, green, cyan, and blue QD-LED devices by assuming that two QD layers are stacked in the EML. The flat-band energy diagram, simulated electric potential distributions, energy distributions, and electric-field distributions for the respective devices are plotted in Supplementary Fig. 5. The simulated hole and electron densities accumulated at the QD layer and the radiative recombination rates per unit area with respect to the applied voltage are also shown in Supplementary Fig. 6. In Fig. 3e, the holes and electrons accumulated at the boundaries of HTL and ETL are injected after a certain threshold voltage due to the electric-field dependent carrier capture process described in the Method section. The accumulation of charge carriers at the QDs abruptly increase as the voltage increases after the threshold voltage. In the simulation, even though the red QD-LED shows a weaker electric-field in the QD layer than the others due to their larger QD particle sizes, the amount of accumulated holes and electrons in the QD layer of the red device is comparable to the other devices due to it having the largest SRH lifetime compared to the other QDs (Supplementary Table 3).

Figure 3f–h show the simulated voltage dependencies of current density, luminance, and EQE values of the red, green, cyan, and blue QD-LED devices for their respective peak wavelengths of 616 nm, 559 nm, 502 nm, and 450 nm which are theoretically obtained by the colour optimisation process. The luminance of the red, green, cyan, and blue QD-LEDs at 7 V were calculated to be 24,877 cd m$^{-2}$, 11,965 cd m$^{-2}$, 6573 cd m$^{-2}$, and 190 cd m$^{-2}$, respectively (Fig. 3f). The threshold voltages of the simulated red, green, cyan, and blue QD-LED devices were 2.0 V, 2.4 V, 3.0 V, and 3.7 V (Fig. 3f). The threshold voltages are determined by the energy-band-offset between the QD layer and the transport layer, according to the electric-field dependent charge injection model[30]. The band-offset increases as the optical bandgap of QDs increases, and the threshold voltage increases in the order of red, green, cyan, and blue QD-LEDs. The maximum EQEs are observed to be the same as the experimental results since the material-level parameters are obtained to match the simulated EQE-current density curves to the experimental EQE-current density curves of the fabricated devices (Fig. 3g–h). The simulated EQE, luminance and current density curves coincide very well with the experimental results for all the QD-LED devices. The simulation presented is successful in accurately predicting the emission spectra of the monochromatic devices using the charge transport model based on electric-field dependent charge injection process coupled with material-level parameters extracted from the experiment.

**Layout design of white lighting system and experimental validation.** In this section, the layout of the pixelated emission patterns of the four QD-LEDs in the 4-primary white lighting system is designed to obtain the emission spectrum optimised by the colour optimisation process. Figure 4a shows the spectral radiances of monochromatic red, green, cyan, and blue QD-LEDs simulated by the charge transport model. The red QD-LED shows the highest peak owing to its highest PLQY QD layer among the QDs (Supplementary Table 3). First, to compare the performance improvement by the optimisation of the emission pattern widths **w** of the multiple pixelated QD-LEDs, a 4-primary lighting system with identical emission pattern widths is taken as a reference. The simulated spectral radiances of the 4-primary lighting system having identical QD pattern widths are shown in Fig. 4b. The spectral radiance of the 4-primary lighting system is derived by summing the simulated four monochromatic spectra of

individual QD-LEDs with the weighting factors related to the emission pattern widths of the pixelated QD-LEDs. The weighting factors of four QD patterns are equally set to be 0.25 in case of identical emission widths (Inset in Fig. 4b).

Figure 4c shows the simulated spectral radiances at different voltages for the 4-primary lighting system having different emission pattern widths designed by the charge transport simulation. Each of the weighting factors $a_n$ for $n$-th QD pattern is obtained by Eq. (2) for the respective peak radiance $A_n$ of the $n$-th monochromatic QD-LED at 5 V and the normalised emission power $\beta_n$ predetermined by the colour optimisation process.

$$a_n = \frac{\beta_n}{A_n} \left[ \sum_{i=1}^{N} \frac{\beta_i}{A_i} \right]^{-1} \tag{2}$$

With the given weighting factors, the emission width $w_n$ of the $n$-th pixelated QD pattern is calculated by $w_n = a_n \times P$ for the horizontal pitch $P$ of the QD pixel group. By applying the emission widths determined by the weighting factors, an adjusted peak radiance $A_n^*$ of $n$-th pixelated QD-LED in the white lighting system is obtained by $A_n^* = a_n \times A_n$ (Fig. 1j). From the design rule in Eq. (2), the weighting factors for the optimised lighting system are determined to be 0.017, 0.118, 0.130, and 0.735 for the red, green, cyan, and blue QD patterns to have the emission spectrum satisfying the maximum CRI and D65 chromaticity at the applied voltage of 5 V (Inset in Fig. 4c).

The experimental EL spectra of the fabricated monochromatic QD-LED devices are given in Fig. 4d for comparison with the simulation results. The EL peak wavelengths of red, green, cyan, and blue are measured to be 626 nm, 563 nm, 506 nm, and 456 nm, respectively. There are slight differences in the peak wavelengths between PL and EL spectra caused by the quantum-confined Stark effect of an external electric-field upon the emission spectrum in QDs[47]. To validate our simulation results, 4-primary lighting systems with identical and optimised QD patterns were fabricated using the transfer printing technique[24–26]. The horizontal pitch $P$ of the QD pixel group consisting of four QD patterns is 375 μm and the total size of the actual lighting system is $3.0 \times 1.5$ mm$^2$ with eight QD pixel groups of stripe red, green, cyan and blue QD patterns. The horizontal pitch $P$ was determined by considering two aspects of (i) the human perception of the pixel group and (ii) the dependency of the pixel pitch on the EL performance in our experiments. First, by selecting the horizontal pitch to be 375 μm, the pixel group was designed not to be recognised by the human visual system at a typical distance from the lighting environment (Supplementary Fig. 7). Second, from our experiment, we selected 375 μm horizontal pitch since the lighting system with 375 μm pitch showed higher EL performance compared with the lighting system with 1500 μm (Supplementary Fig. 8). Figure 4e shows the spectra of the fabricated lighting system with the identical emission widths of 93.75 μm (for the weighting factors of 0.25). The spectrum of the fabricated lighting system having the optimised emission widths is plotted in Fig. 4f. The optimal emission withs for red, green, cyan, and blue QD patterns are designed to be 6 μm, 44 μm, 49 μm, and 276 μm for the respective optimised weighting factors of 0.017, 0.118, 0.130, and 0.735. Insets in Fig. 4e, f are the snapshots of the fabricated white lighting systems with identical and optimised emission widths under EL-driving operation. After the optimisation of the emission width, the blue pixelated QD-LED exhibits a maximum radiance while the red pixelated QD-LED shows a minimum as shown in Fig. 4f.

Figure 4g–i show the simulated electro-optical properties of the hypothetical 4-primary QD-LED lighting systems finally optimised by our computational design framework. The luminances

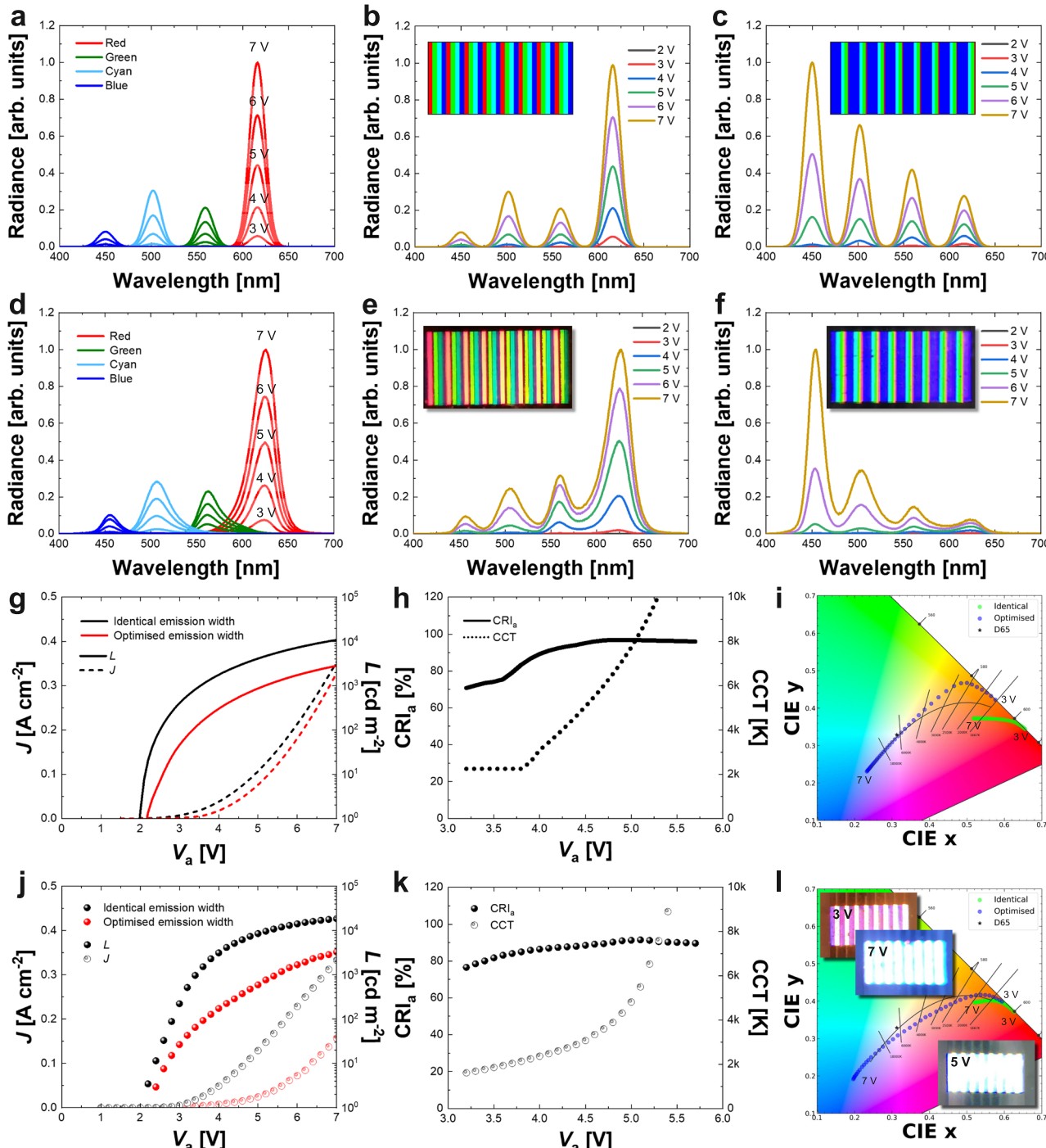

**Fig. 4 Layout design of 4-primary white lighting system architecture and validation of computational design framework by the experiments.**
**a** Simulated spectra of the monochromatic red, green, cyan, and blue QD-LEDs for the voltages from 3 V to 7 V with 1 V step. Simulated spectra of the 4-primary lighting systems having **b** identical and **c** optimised QD emission pattern widths for the voltages from 2 V to 7 V with 1 V step. Insets are the layouts of the pixelated QD-LEDs for the identical (**b**) and the optimised (**c**) emission pattern widths. **d** Experimental spectra of the fabricated monochromatic red, green, cyan, and blue QD-LED devices for the voltages from 3 V to 7 V with 1 V step. **e, f** Spectra of the fabricated 4-primary lighting systems having **e** identical and **f** optimised QD emission pattern widths for the voltages from 2 V to 7 V with 1 V step. Insets are the snapshots of EL-driven white lighting systems with the identical (**e**) and the optimised (**f**) emission pattern widths. **g** Voltage dependencies of current density (*J*) and luminance (*L*) curves of the lighting systems calculated from the simulation. **h** Simulated average CRI (CRI$_a$) and the correlated colour temperature (CCT) variations of the lighting system having optimised emission widths. **i** Voltage-dependent chromaticity variations of the lighting systems having identical (green dots) and optimised (blue dots) obtained from the simulation. The chromaticity of the CIE standard illuminant D65 is marked with black star. **j** Experimental voltage dependencies of current density and luminance curves for the fabricated lighting systems. **k** Voltage dependencies of the CRI$_a$ and CCT for the fabricated lighting system having the optimised emission widths. **l** Voltage-dependent chromaticity variations of the fabricated lighting systems having identical (green dots) and optimised (blue dots) emission pattern widths.

of the lighting systems with identical and optimised emission widths were simulated to be 10,902 cd m$^{-2}$ and 2824 cd m$^{-2}$ at 7 V, respectively (Fig. 4g). The theoretical CRI varies from 83% to 97% and the CCT varies linearly from 2243 K to 9207 K for the voltage ranging from 3.8 V to 5.2 V (Fig. 4h and Supplementary Fig. 9). Figure 4i shows the simulated colour loci of the white lighting system for the applied voltage. In the simulation, the colour of the optimised lighting system changes widely from red to blue through white when the voltage changes from 3 V to 7 V, whereas the system with identical emission widths hardly changes its colour from red even for the voltage increment. As a result of the computational design, excellent lighting performances of 97% CRI and chromaticity close to D65 illuminant around 5 V with a wide colour variation have been theoretically achieved by optimising the colour combination of QDs and designing the layout of the pixelated emission patterns of QDs.

The changes in the colour chromaticity are explained by the variation of the peak radiances for each QD with respect to the applied voltage. The theoretical peak radiances obtained at peak wavelengths of red (616 nm), green (559 nm), cyan (502 nm), and blue (450 nm) QDs with respect to the applied voltage are plotted in Supplementary Fig. 10 for the simulated lighting systems having identical and optimised emission weighting factors. Since each of the QD patterns is turned on in the order of red, green, cyan, and blue due to their different threshold voltages, the colour chromaticity can be changed by the applied voltage. In the reference lighting system with identical emission widths, the red emission always dominates other colours over the entire voltage range (Supplementary Fig. 10a), and the colour chromaticity of the reference lighting system barely changes its colour from red. Meanwhile, the optimised white lighting system shows a wide chromaticity change with the voltage due to the change of voltage dependencies of emission curves caused by the new layout design of the pixelated QD patterns (Supplementary Fig. 10b).

Figure 4j–k show the experimental electro-optical properties of the fabricated 4-primary QD-LED lighting systems. The luminances of the devices with identical and optimised emission widths are measured to be 18,540 cd m$^{-2}$ and 3285 cd m$^{-2}$ at 7 V, respectively (Fig. 4j). The fabricated lighting system having optimised emission widths shows the variation of average CRI from 77% to 92% and CCT from 1612 K to 8903 K for the voltage varying from 3.2 V to 5.4 V (Fig. 4k and Supplementary Fig. 9). The white QD-LED lighting system also shows the EL performances of 1.4% EQE and 2.4 cd A$^{-1}$ current efficiency which are comparable to the values from the previous literature on white QD-LED lighting (Supplementary Fig. 11 and Supplementary Table 2).

Figure 4l shows the colour loci of the fabricated white lighting system for the applied voltage varying from 3 V to 7 V. In Fig. 4l, the fabricated white lighting system with optimised emission widths also shows very wide colour changes from red to blue through white as explained above and in Supplementary Fig. 10. Meanwhile, the fabricated lighting system with identical emission widths hardly changes its colour from red over the voltage due to the remarkably large emission contribution of the red QD pattern under the identical emission widths. Insets in Fig. 4l are the snapshots of the fabricated lighting system emitting reddish, white, and bluish lights at the applied voltages of 3 V, 5 V, and 7 V, respectively. Owing to the wide colour controllability, the optimised white lighting system is able to respond to most of the colour preferences affected by the circadian rhythm for daily life (temporal difference), region/culture (spatial difference), or lighting context (situational difference)[48–52]. In addition, the colour locus of the optimised white lighting system traces close to the Planckian locus over the entire voltage range, enabling it to express a more precise colour close to the specified CCT

(Supplementary Fig. 12). Together with the high CRI and wide CCT variation, the colour accuracy of our optimised white lighting system for the specified CCT would motivate discourse toward establishing industry standards for smart lighting.

The architecture of the optimised lighting system proposed and verified through experiment here shows excellent lighting properties for next-generation versatile smart lighting, providing accurate colour rendering while also allowing wide colour controllability with high colour precision. The experimental results of the fabricated white lighting coincide very well with the simulation results from our computational design framework, especially for the emission spectrum, CRIs, and loci of colour variation in the CIE colour coordinates with respect to the driving voltage. In summary, the methodology of using a computational design framework which includes a combinatorial colour optimisation process using the Nelder-Mead algorithm and complete charge transport simulation using the electric-field dependent charge injection model is very appropriate for designing and predicting the electro-optical properties of multi-primary white lighting systems based on pixelated QD-LEDs.

## Discussion

In this work, a computational design framework has been proposed for optimising the architecture of the smart white lighting systems having a layout of the multiple patterns of pixelated QD-LEDs. The computational design framework consists of system-level design through a combinatorial colour optimisation process and the layout design of multiple pixelated QD-LEDs based on rigorous device-level charge transport simulation. For the system-level design, the number of primary colours, its peak wavelengths and emission peaks were optimised efficiently by a combinatorial colour optimisation process using the Nelder-Mead method to achieve high CRI with a white chromaticity close to CIE D65 illuminant. The electro-optical properties including the emission spectra of individual QD-LED devices were analysed using device-level charge transport simulation with an electric-field-dependent charge injection model. The material-level parameters used for the charge transport simulations are extracted from measurements on device fabrication and TEM analysis. Using the simulated emission spectrum at a given voltage, the layout of the QD emission pattern widths was designed to match the emission spectrum optimised by the colour optimisation process. The simulated 4-primary white lighting system with red (616 nm), green (559 nm), cyan (502 nm), and blue (450 nm) exhibit a theoretical maximum of 97% average CRI with a CCT variation from 2243 K to 9207 K for the voltage ranging from 3.8 V to 5.2 V.

To experimentally confirm the simulation results, an optimised 4-primary white lighting system having multiple pixelated patterns of red, green, cyan, and blue QD-LEDs was fabricated. The pixelated emission patterns of individual QD-LED were obtained using the transfer printing technique. The EL peak wavelengths of the fabricated red, green, cyan, and blue QD-LEDs are measured to be 626 nm, 563 nm, 506 nm, and 456 nm, respectively. The fabricated lighting system also exhibits an excellent averge CRI of 92% with a wide CCT variation from 1612 K to 8903 K for the driving voltage ranging from 3.2 V to 5.4 V. The experimental results of the fabricated white lighting system coincide very well with the simulation results obtained from our computational design framework, especially for the emission spectrum, CRIs and colour loci in the CIE colour coordinates with respect to the driving voltage. In conclusion, the pixelated QD-LED based white lighting system optimised by our computational design frame-work shows an excellent colour rendering capability for clear visual information and wide colour controllability from red to

blue which can accommodate personal mood changes and circadian rhythm in an ambient living environment.

## Methods

**Colour optimisation process**. To find the optimal combination of peak wavelengths for the $N$-primary coloured white lighting system, we used a combinatorial colour optimisation process with a cost function for a given spectral radiance of the lighting system. First, the spectral radiance $L(\lambda, \Lambda)$ for a set of peak wavelengths $\Lambda = (\lambda_1, \lambda_2, ..., \lambda_N)$ of the $N$-primary white light is defined as a mixture of Gaussian-shaped spectral radiances for the $n$-th primary colour with peak wavelength $\lambda_n$, normalised power $\beta_n$, and its full-width at half maximum (FWHM) $\Delta\lambda_n$.

$$L(\lambda, \Lambda) = \sum_{n=1}^{N} \beta_n \exp\left[-\frac{4\ln2}{\Delta\lambda_n^2}(\lambda - \lambda_n)^2\right] \quad (3)$$

Here, $\beta_n = D65(\lambda_n)$ for the normalised power distribution of the CIE standard illuminant D65. With the given spectral radiance $L(\lambda, \Lambda)$, we calculate a cost function that is a function of the CRIs and chromaticity difference $\Delta_{xy}$ according to Eq. (1). In Eq. (1), the colour properties such as $\mathbf{CIE}_{xy}$ and CRIs for the spectral radiances are calculated with Colour package which is an open-source library for Python[53,54]. The factor 1.5 of the first term in Eq. (1) encourages the optimiser to prioritise chromaticity over CRI. With the given cost function, the colour optimisation process simultaneously minimises chromaticity difference while maximising the CRI of the $N$-primary white lighting system.

In the combinatorial colour optimisation process, we used an iterative procedure with the Nelder-Mead algorithm which finds the local minimum point and value of the given cost function in an $N$-dimensional variable space of the peak wavelength set $\Lambda$. The optimised wavelength set $\Lambda^{\text{opt}}$ is determined to be a wavelength set having the smallest cost function, by the end of the iterative procedure, as described in Eq. (4).

$$\Lambda^{\text{opt}} = \arg\min_m f_m^{k+1}(\Lambda_m^{k+1}) \quad (4)$$

Here, $\arg\min f(x_m)$ is the $x_m$ that gives smallest $f$ for $m = 1, 2, ..., N + 1$. A programming function of the Nelder-Mead algorithm supported by TensorFlow, which is an open-source Python package for machine learning frameworks, was utilised in this study[33,55].

**Charge transport model in QD-LED device**. The injection current densities of the hole and electron $J_p^i$ and $J_n^i$ are described by an electric-field-dependent carrier capturing process between the QD layer and charge transport layers. The injection current densities are described by Eq. (5)[30].

$$J_p^i = \frac{1}{2}qd_{QD}s_{QD}\sigma_p T_p \mu_p^{QD} F_p p_h(N_{QD} - p_{QD}),$$
$$J_n^i = -\frac{1}{2}qd_{QD}s_{QD}\sigma_n T_n \mu_n^{QD} F_n n_e(N_{QD} - n_{QD}) \quad (5)$$

Here, $q$ is the electric charge of a proton, and $d_{QD}$ and $s_{QD}$ are the diameter and the cross-sectional area of the QDs, respectively. The relative capture cross-sections are given by $\sigma_p$ and $\sigma_n$, with $T_p$ and $T_n$ being the hole and electron tunnelling probabilities of the energy barrier built by the shell covering QD core. The hole and electron densities at the HTL and ETL surfaces facing QD layer are given by $p_h$ and $n_e$, respectively. $p_{QD}$ and $n_{QD}$ are the hole and electron densities at the centre of the QDs adjacent to the HTL and ETL, and $N_{QD}$ is the density of QDs calculated by $N_{QD} = (3/4\pi) \times (d_{QD}/2)^{-3}$. $\mu_p^{QD}$ and $\mu_n^{QD}$ are hole and electron mobilities in QD layer and are assumed equally to be $2.6 \times 10^{-5}$ cm$^2$ V$^{-1}$ s$^{-1}$ in the simulation. $F_p$ and $F_n$ are the electric-field intensities acting on holes and electrons at the interfaces between QD and the HTL or ETL, respectively.

The drift-diffusion current densities of the hole and electron in the transport layers, $J_p^d$ and $J_n^d$, are described by summation of the drift and diffusion current densities as expressed in Eq. (6)[29,35].

$$J_p^d = q\mu_p pF_p - \mu_p k_B T\frac{\partial p}{\partial z}, \quad J_n^d = q\mu_n nF_n + \mu_n k_B T\frac{\partial n}{\partial z} \quad (6)$$

Here, $\mu_p$ and $\mu_n$ are the hole and electron mobilities in the given semiconductors, $F_p$ and $F_n$ are the electric fields acting on holes and electrons across the transport layers. $k_B$ is the Boltzmann constant and $T$ is the absolute temperature of the device. The electric fields in Eqs. (5) and (6) are calculated by the gradient of the energy-level distribution of hole and electron across the device. The energy-level distribution is the summation of the flat-band energy-level of the carriers and the potential distribution which is solved by Poisson's equation for the given carrier densities and the boundary conditions of a bias voltage configuration.

The tunnelling current densities, $J_p^t$ and $J_n^t$ from the $m$-th QD to $(m + 1)$-th QD, are expressed as Eq. (7)[27,30].

$$J_p^t = -q\zeta_p d_{QD}(p_{m+1} - p_m), \quad J_n^t = q\zeta_n d_{QD}(n_{m+1} - n_m) \quad (7)$$

Here, $\zeta_p$ and $\zeta_n$ are the tunnelling coefficients of hole and electron between two neighbouring QDs. The tunnelling coefficients can be described by the following equation[30];

$$\zeta = \frac{\sqrt{2k_B T/m_c^*}}{d_{QD}}\exp\left(-\frac{2d_s}{d_t}\right) \quad (8)$$

where $m_c^*$ and $d_{QD}$ are the effective mass of carriers in the core of QDs and the diameter of QDs. $d_s$ is the shell thickens of QD and $d_t$ is the tunnelling distance expressed as $d_t = h/(2\pi) \times (8 m_s^* \Delta E)^{-0.5}$ with the plank constant $h$ and the effective mass $m_s^*$ of the QD shell. $\Delta E$ is the energy-band barrier height between the core and shell of QDs. The detailed material-level parameters for the shell material is described in Supplementary Table 1.

The simulation model uses continuity equations as the governing equations for the calculation of dynamic behaviours of holes and electrons and Poisson's equation for solving the electrostatic potential across the QD-LED device[27,28,30,35,56]. As a numerical technique for solving the second-order partial differential equations of the continuity equations combined with Poisson's equation, we used the finite difference method (FDM) on a discretised grid space across the QD-LED device[57].

The bimolecular radiative recombination rate $U_{RAD}$ at the centre of the QDs is calculated from Eq. (9) with the simulated hole and the electron densities at the centre of QD layers.

$$U_{RAD} = \gamma(p^{QD} n^{QD} - n_i^2) \quad (9)$$

Here, $\gamma$ is the Langevin radiative recombination strength which is defined as $\gamma = q(\mu_p^{QD} + \mu_n^{QD})/(\varepsilon_r^{QD}\varepsilon_0)$ with the hole and electron mobilities $\mu_p^{QD}$ and $\mu_n^{QD}$, dielectric constant of the QD layer $\varepsilon_r^{QD}$ and the vacuum permittivity $\varepsilon_0$. In the simulation, $\gamma$ is set to be $1.0 \times 10^{-11}$ cm$^3$ s$^{-1}$ for $\mu_p^{QD} = \mu_n^{QD} = 2.6 \times 10^{-5}$ cm$^2$ V$^{-1}$ s$^{-1}$ and $\varepsilon_r^{QD} = 9.4$. $n_i$ is an intrinsic carrier concentration of the QD core material. The SRH recombination rate $U_{SRH}$ and the Auger recombination rate $U_{AUG}$ at the centre of the QDs are also used for non-radiative recombination processes in the device as shown in Eqs. (10) and (11)[35,56].

$$U_{SRH} = \frac{pn - n_i^2}{\tau(p + n)} \quad (10)$$

$$U_{AUG} = C(p + n)(pn - n_i^2) \quad (11)$$

Here, $\tau$ is the SRH recombination lifetime and $C$ is the Auger capture probability, respectively.

The EQE of the device under a given bias condition is calculated from the simulated radiative and non-radiative recombination rates at each of the QD centre. By integrating each of the recombination rates $U_{RAD}$, $U_{SRH}$, and $U_{AUG}$ over the entire region of the QD-LED device, each of the recombination rate per unit area $R_{RAD}$, $R_{SRH}$, and $R_{AUG}$ for the Langevin radiative recombination, the SRH and Auger non-radiative recombinations can be calculated. Finally, the EQE of the QD-LED device can be obtained by Eq. (12) with the PLQY of QD layer $\eta_{QD}$ for the respective QDs and the light extraction efficiency $\eta_{ext}$ of 0.2.

$$EQE = \frac{\eta_{QD} \times \eta_{ext} \times R_{RAD}}{R_{RAD} + R_{SRH} + R_{AUG}} \quad (12)$$

For a given Langevin radiative recombination strength $\gamma$, the PLQY of the QD layer $\eta_{QD}$, the SRH recombination lifetime $\tau$, and the Auger capture probability $C$ are determined to match the simulated EQE-current density curves to the experimental EQE-current density curves of the fabricated QD-LED devices (Supplementary Table 3).

**Derivation of maximum EQE for QD-LED device**. In Eq. (12), the EQE is mainly determined by the PLQY of QD layer, Langevin radiative recombination strength $\gamma$, SRH recombination lifetime $\tau$, and Auger capture probability $C$ of the QDs, and is a function of the charge densities $p^{QD}$ and $n^{QD}$ of the hole and electron accumulated in the QD layer for a given applied voltage. Equation (12) can be simplified to Eq. (13) under the assumption of the charge-balanced condition $(p^{QD} = n^{QD})$[58,59].

$$EQE = \eta_{ext}\eta_{QD}\frac{\gamma k}{1/(2\tau) + \gamma k + 2Ck^2} \quad (13)$$

Here, $\eta_{ext}$ is an optical light extraction efficiency related to the multi-layered structure of the QD-LED device, and $\eta_{QD}$ is the PLQY of the given QD layer. $k$ denotes the charge carrier density at the centre of QDs, which is a function of the applied voltage. Under the assumption of charge-balanced condition, $k$ equals $p^{QD}$ or $n^{QD}$. Then, the maximum EQE of a given QD-LED device is determined by Eq. (14) derived from $d(EQE)/dk = 0$.

$$EQE_{max} = \eta_{ext}\eta_{QD}\frac{Q}{2 + Q} \quad (14)$$

Here, a quality factor $Q = \gamma \times (\tau/C)^{1/2}$ is introduced, which is determined by the Langevin radiative recombination strengths $\gamma$, SRH recombination lifetime $\tau$, and the Auger capture probability $C$. Since the $EQE_{max}$ is directly proportional to the PLQY $\eta_{QD}$ and also is a function of the quality factor $Q$, the larger the quality factor and higher the PLQY of the QD-LED device, the higher $EQE_{max}$ can be obtained.

**Determination of QD emission pattern width.** The spectral radiance of the monochromatic QD-LED device is directly calculated from the radiative recombination rate per unit area $R_{RAD}$ of the device. For the given peak wavelength $\lambda_n$ of the $n$-th primary colour, where $n$ is a natural number from 1 to $N$, the spectral radiance $S(\lambda, \lambda_n)$ is obtained from Eq. (15) with the assumption of the Gaussian distribution for wavelength $\lambda$.

$$S(\lambda, \lambda_n) = A_n \exp\left[-\frac{4 \ln 2}{\Delta \lambda_n^2}(\lambda - \lambda_n)^2\right] \quad (15)$$

Here, $A_n$ is the peak radiance of the $n$-th primary QD-LED device under a given applied voltage and $\Delta \lambda_n$ is the FWHM of the given QD-LED device. $A_n$ is expressed as Eq. (16) with the PLQY of the $n$-th QD layer $\eta_n$ and the radiative recombination rate per unit area $R_n$.

$$A_n = \frac{2\sqrt{\ln 2}}{\Delta \lambda_n \pi \sqrt{\pi}} \frac{hc}{\lambda_n} \times \eta_n \times \eta_{ext} \times R_n \quad (16)$$

Here, $c$ is the speed of light and $h$ is the Planck's constant. The total spectral radiance $L(\lambda, \boldsymbol{\Lambda})$ of the white lighting system with the set of weighting factors $\mathbf{a} = (a_1, a_2, ..., a_N)$ for the set of $N$-primary pixelated QD-LED patterns is obtained by a weighted summation of individual spectrum $S(\lambda, \lambda_n)$ as described in Eq. (17).

$$L(\lambda, \boldsymbol{\Lambda}) = \sum_{n=1}^{N} a_n S(\lambda, \lambda_n) \quad (17)$$

Here, $a_n$ is a weighting factor for the emission widths of $n$-th primary QD pattern, hence the summation of $a_n$ for all the primary colours is always equal to 1. For the $N$-primary coloured white lighting system having $N$ number of QD patterns with identical emission widths, $a_n$'s are equally given to be $1/N$. However, for the lighting system having QD patterns with different emission widths, each weighting factor $a_n$ is determined from Eq. (2) for the peak radiance $A_n$ of the monochromatic device and the predetermined normalised power $\beta_n$ at the given peak wavelength $\lambda_n$ for $n$-th primary QD-LED in the white lighting system.

**Device fabrication and measurements.** To compare the simulation results with experiments, red, green, cyan, and blue QD-LEDs were fabricated by the transfer printing technique. An ITO, a PEDOT:PSS, and a TFB were used for the anode, HIL and the HTL, respectively. A MZO and an Al were used for the ETL and the cathode electrode. The CdSe/ZnS core/shell red, green, cyan, and blue QDs having the respective PL peak wavelengths of 621 nm, 558 nm, 502 nm, and 452 nm were purchased from Xingshuo Nanotech Co., Ltd. to form EMLs of each QD-LED devices. The solution phase of PEDOT:PSS (Al 4083), TFB (8 mg ml$^{-1}$ in chlorobenzene), MZO (70 mg ml$^{-1}$ in butanol) were used for the materials of HIL, HTL and ETL, respectively. After cleaning the patterned ITO glass substrate, PEDOT:PSS layer was spin-coated at 3000 revolutions per minute (rpm) spin-speed and annealed at 150 °C on the hot plate for 30 minutes. Subsequently, TFB precursor was spin-coated at 3,000 rpm spin-speed and annealed at 130 °C on the hot plate for 30 minutes.

The QD layer was formed on the HTL by the transfer printing technique. The fabrication process of the patterned QD-LEDs via the transfer printing technique is illustrated in Supplementary Fig. 13. The QDs for the individual primary colour were picked up by stripe-patterned polydimethylsiloxane (PDMS) stamp from a self-assembled monolayer-treated donor substrate and transferred onto the pre-processed HTL with a transfer printing equipment. The micro-patterns of PDMS stamps were replicated from a master mould made of photoresist on the silicon wafer through the photolithography process. The pattern width and the pitch of the individual QD layer were duplicated by the width and pitch of the micro-patterns on PDMS stamps. By repeating these pick-up and transfer printing processes for all the QDs, each of the patterned QD layers is formed at spatially separated locations within a single lighting system.

The MZO layer was sequentially formed on the QD layer by spin-coating under the spin-speed of 3000 rpm and annealed at 100 °C on a hot plate for 10 minutes. Finally, the aluminium layer was deposited on the MZO surface to form a cathode electrode via a thermal evaporation process. The electro-optical characteristics of current densities, lumiance, and EQE were obtained by the External quantum efficiency measurement system C9920-12 from Hamamatsu Photonics K.K.

## Data availability

The source data generated in this study are provided in the Source Data file. Source data are provided in this paper.

## Code availability

The code for the colour optimisation process is available on GitHub (https://github.com/chaturatbs/nst-qd-white-lighting). The code for charge transport simulation is available from the corresponding authors upon reasonable request. All requests will be reviewed for intellectual property or confidentiality.

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

## Acknowledgements
This research was supported by the UK Engineering and Physical Sciences Research Council (EPSRC) project EP/P027628/1 'Smart Flexible Quantum Dot Lighting' and by the European Union under H2020 grant agreement No 685758 '1D-NEON'. C.S. acknowledges support from the EPSRC through the doctoral training partnership (DTP) scheme, studentship award EP/N509620/1.

## Author contributions
C.S. performed the colour optimisation. H.W.C., S.L., D.-W.S., S.Y.B. and J.-W.J. fabricated the devices. X.-B.F. and L.N. measured the material properties. J.Y. and Y.K. measured the device performances. S.-M.J. conceived the idea and performed the charge transport simulation. L.G.O. designed the experiments. G.A. and J.M.K. supervised this work. C.S., H.W.C. and S.L. contributed equally. All the authors discussed the results and contributed to the preparation of the manuscript.

## Competing interests
The authors declare no competing interests.
