## [Peer Review File · Nature Communications]

Optoelectronic System and Device Integration for
Quantum-Dot Light-Emitting Diode White Lighting with
Computational Design FrameworkREVIEWER COMMENTS

Reviewer #2 (Remarks to the Author):

The manuscript entitled "Optoelectronic System and Device Integration for Quantum-Dot Light Emitting Diode White Lighting with Computational Design Framework" by Sung-Min Jung is very interesting in the way that the proposed white light system using multiple pixelated QD-LEDs would be very promising for our daily life ambient lighting system. The authors explained in a very well manner about the optimization of color rendering and the broad emission wavelengths of the device. The simulation study and the experimental results are in good agreement and the authors also added a detailed explanation about the charge transport model in the device, which will be quite interesting for the readers as well. This manuscript will give a significant impact on the QD-LED field and it is deserved to be published in Nature Communications. However, I do have some concerns which are listed as follows.

1. How about the stability of QDs and the stamp printing method?
2. The authors are encouraged to put an explanation in the manuscript why the individual colors have different EQE even though it is obvious from the Figure, to give a clear understanding to the readers and make a better manuscript.
3. Since the performance of the white light is deteriorated by low EQE colors, so what optimization can be done to improve the EQE for those colors.
4. The authors are encouraged to put an explanation about the change of color chromaticity with applied voltage.

Reviewer #3 (Remarks to the Author):

Samarakoon et al. proposed a computational design framework for the first time to design the architecture of a white lighting system having a layout of multiple pixelated patterns of electric-field-driven quantum dot light-emitting diodes (QD-LEDs). The fabricated white lighting system using the transfer printing technique exhibits excellent lighting performances of 92% CRI and wide color temperature variation from 1612 K to 8903 K with only four pixelated QD-LEDs. This strategy of computational design can be targeted to direct the fabrication of the efficient and pixelated white QD-LEDs, and the work is novel. However, I think there still exists a few of problems:

- (1) Is this computational design just suitable for the fabrication of pixelated white QD-LEDs. Based on this method, whether the computational design can be further developed as to white QD-LEDs via other fabricating technology. And what does need to modify in this method.

(2) Could the EL performance of white QD-LEDs in this work be provided? Where does this work stand in, compared with other white devices reported by literatures?

(3) Taking 4-primary colors for an example, as shown in Figure 4e, how to fabricate the high-quality white QD-LEDs with the identical emission widths? Namely, with the identical emission widths of 93.75 μm , which requirements do the corresponding QD materials need to satisfy based on the computational design method, and to realize for smart lightings. If the smart lightings can be realized based on the identical emission widths, what's the difference between white QD-LEDs with the identical emission widths and that with the optimised emission widths?

(4) Why is 375 μm selected as the horizontal pitch P of the QD pixel group consisting of four QD patterns? Whether the pixel size can be further reduced, maybe the higher the pixel, the more useful for pixelated QD-LEDs? The EL performance maybe related to the pixel size of devices. When the pixel size is reduced, how the EL performance of white QD-LEDs will be show? Whether this or similar computational design methodology can be targeted to direct the relation of pixel size to the device performance?

(5) Smart lightings are expected to respond to the personal mood or circadian rhythm in the daily ambient living while also rendering colours of objects more accurately. Except that, I want to know more detailed requirements for the smart lightings. What's more, whether the discrepancy is still significant between the results in this work and the standards for smart lightings?

Author's Response Letter

7 April 2022

Dear Reviewers,

Thank you sincerely for taking the time to evaluate our manuscript entitled "***Optoelectronic System and Device Integration for Quantum-Dot Light-Emitting Diode White Lighting with Computational Design Framework***" (NCOMMS-21-43332). We would like to submit this author's response letter to point-by-point response to the comments that reviewers had raised. We truly appreciate your in-depth review and valuable comments on our manuscript.

We tried to accommodate the reviewer's comments as much as possible and elaborately revised the manuscript with crystal-clear responses. In this response letter, the reviewer's comments are in bold and italic, the author's responses are in plain text, and the corrections are in red text. We also have included all changes in the revised manuscript and supplementary information with red coloured texts. Thanks to the valuable comments from the reviewers, we think that the revised manuscript has been much improved from the original manuscript. We truly hope that our revised manuscript will fit the reviewer's expectations.

On behalf of all authors, yours sincerely,

Sung-Min Jung, PhD
Senior Research Associate
Dept of Engineering
University of Cambridge

Reviewer #2

The manuscript entitled "Optoelectronic System and Device Integration for Quantum- Dot Light Emitting Diode White Lighting with Computational Design Framework" by Sung-Min Jung is very interesting in the way that the proposed white light system using multiple pixelated QD-LEDs would be very promising for our daily life ambient lighting system. The authors explained in a very well manner about the optimization of colour rendering and the broad emission wavelengths of the device. The simulation study and the experimental results are in good agreement and the authors also added a detailed explanation about the charge transport model in the device, which will be quite interesting for the readers as well. This manuscript will give a significant impact on the QD-LED field and it is deserved to be published in Nature Communications. However, I do have some concerns which are listed as follows.

Response:

Thank you sincerely for taking the time and effort in evaluating our manuscript and for the positive remarks with valuable comments. As we understand, reviewer #2's comments are related to the device performances, such as the stability of the device, difference of EQEs for individual QD-LED devices, ideas for improving EQEs of transfer-printed QD-LEDs, and the mechanism of the colour change in our white lighting system. These are big topics in the development of QD-LED devices, which should be dealt with in detail by further research. We tried our best to address all the questions about the stability, EQEs as well as the explanation of the colour change mechanism. We hope that you will find the improvement in the revised manuscript in accordance with your valuable review and comments.

Reviewer #2-(1) *How about the stability of QDs and the stamp printing method?*

Response:

Thank you for the reviewer's insightful question on the stability of the QDs and the stamp printing method. The white lighting system proposed in this study is composed of multiple pixelated QD patterns, and each QD pattern is fabricated by transfer printing technology using a polydimethylsiloxane (PDMS) stamp. Regarding the reviewer's question, the authors tried to explain the stability of QDs and the transfer printing method in this letter, by investigating the device lifetimes of QD-LEDs reported in recent publications and by experimentally testing the lifetime of our fabricated QD-LED device.

The device lifetime of QD-LEDs is directly affected by the stability of the QDs. Since the core of the QDs is degraded by reaction with oxygen and water molecules, it is necessary to protect the QD core nanocrystal from surface oxidation to enhance the stability of QDs¹. Based on the robust design of the materials and structures of the shell layer and surface-binding ligands protecting the QD core crystal, many of research has been performed to enhance the device lifetime of the QD-LEDs. Among them, notable publications on the QD stabilities in terms of the device lifetime are summarised and listed in Table R1. From the table, the lifetimes T_{50} (time to reach 50% of an initial luminance) of red and green QD-LEDs have already been reached more than 100,000 hours by a spin coating process, which are acceptable in the display and lighting industry. In the case of spin-coated blue QD-LED showing 10,000 hours of lifetime, further improvement on stability is required. Even though enough lifetime of QD-LED is achievable by the spin coating process, it is impossible to fabricate the pixelated QD-LED white lighting system with only the spin coating process.

There are several candidate techniques to fabricate pixelated QD-LEDs, such as photolithography, inkjet printing and transfer printing. The photolithography is the most conventional approach to fabricating fine patterns of various materials. However, the chemicals such as photoresist and developer (solvent) that are used for the lithography process could cause chemical damages to the pre-deposited layers and especially degradation of optical properties of QDs layer^{2,3}. Inkjet printing is attractive for implementing QD patterns, but a non-uniform film with a rough surface morphology of QD layers are still the major challenge to be solved^{4,5}. Transfer printing with an elastic PDMS stamp is one of the promising candidates to stably fabricate pixelated QD patterns without any additional chemical processes that could damage the pre-deposited QD patterns⁶. In this study, the sizes of the QD patterns were

precisely controlled by the transfer printing method to accurately match the optimal emission spectrum designed by the computational design framework. Here, by virtue of the process uniqueness of the transfer printing technique, chemically stable QD pixels could be precisely patterned in separate positions without any damage from other chemicals.

In order to investigate the state-of-the-art device stability of QD-LEDs fabricated by transfer printing, the device lifetimes of the transfer-printed QD-LEDs are also listed in Table R1. From recent publications, the lifetime of the transfer printed QD-LED devices shows maximally 1900 hours for T_{50} at an initial luminance of 100 cd m^{-2} (estimated from the original lifetime of 41.7 hours for T_{50} at an initial luminance of 4554 cd m^{-2} from Choi *et al.*⁷), which is quite shorter than those of the spin-coated QD-LED devices. To experimentally check the stability of the transfer-printed QD-LED device, we fabricated the monochromatic red QD-LED device by the transfer printing process. The device lifetime was tested under very harsh conditions of brightness in the range from $10,000 \text{ cd m}^{-2}$ to $30,000 \text{ cd m}^{-2}$ (Appendix A). Based on our experimental result, T_{50} at an initial luminance of 100 cd m^{-2} is estimated around 40,000 hours which is more than twenty times longer than that of the device reported by Choi *et al.*⁷ (Table R1). The device fabricated by our transfer printing process showed good stability to be comparable to that of the spin-coated devices as well as the world's best stability among the transfer-printed QD-LED devices. Nevertheless, the transfer-printed QD-LED device still needs further improvements in the device lifetime compared to the spin-coated QD-LED devices.

One of the main reasons for the short lifetime of the transfer-printed QD-LED devices is that the QD layer is exposed to the air containing oxygen and moisture during the transfer printing process⁸. Basically, long-term oxygen exposure of QDs induces the formation of surface trap states for permanent fluorescence quenching, resulting in device degradation⁹. Therefore, it is required to block the air containing oxygen or moisture during the transfer printing process. Unfortunately, due to the massive size of the transfer printing equipment in our laboratory, it could not be equipped in an inert gas-filled glove box. Hence the QD layer is inevitably exposed to air containing oxygen and moisture during the transfer printing process. However, this is a laboratory-level problem of our transfer printing equipment, which can be solved by developing oxygen/moisture-blocked transfer printing equipment.

In summary, our transfer-printed QD-LED device shows around 40,000 hours of T_{50} at 100 cd m^{-2} which is the world's best stability among the state-of-the-art transfer-printed QD-LED devices. However, to enhance the stability of the transfer-printed QD-LED devices, further

studies on the stability in terms of QD materials and process technologies should be conducted. If the QD patterning process techniques that enable a much longer lifetime of the device are further developed, they can be also employed in the pixelated white lighting system architecture proposed in this study. According to the reviewer's question, we put explanations of the stability of the QDs and the transfer printing methods below. The changes and additional remarks on the stability of the QDs and the transfer printing methods in the revised manuscript are described below.

Table R1. Summary of QD-LED device lifetimes reported from recent publications.

References	QD film process	QD Colour	Lifetime [hour]	Condition
Yang et al. ¹⁰	Spin coating	Red	90,000	T_{50} at 100 cd m ⁻²
		Green	300,000	T_{50} at 100 cd m ⁻²
		Blue	1,000	T_{50} at 100 cd m ⁻²
Dai et al. ¹¹	Spin coating	Red	100,000	T_{50} at 100 cd m ⁻²
Pu et al. ¹²	Spin coating	Red	90,000	T_{50} at 100 cd m ⁻²
		Blue	10,000	T_{50} at 100 cd m ⁻²
Weiran et al. ¹³	Spin coating	Red	2,200,000	T_{50} at 100 cd m ⁻²
Chen et al. ¹⁴	Spin coating	Red	800	T_{90} at 1000 cd m ⁻²
		Blue	23	T_{50} at 1000 cd m ⁻²
Choi et al. ⁷	Transfer printing	Green	41.7	T_{50} at 4554 cd m ⁻²
			> 1900	T_{50} at 100 cd m ⁻²
Kim et al. ⁶	Transfer printing	Red	76.5	T_{50} at 1050 cd m ⁻²
			> 800	T_{50} at 100 cd m ⁻²
Kim et al. ¹⁵	Transfer printing	Red	35	T_{50} at 2037 cd m ⁻²
			> 700	T_{50} at 100 cd m ⁻²
This work	Transfer printing	Red	40,000	T_{50} at 100 cd m⁻²

Corrections: -----

(line 311, page 13, Main article): Added

Regarding the stability of the transfer-printed device, our fabricated red QD-LED shows a lifetime $T_{50}@100$ cd m⁻² (time to reach 50% of an initial luminance of 100 cd m⁻²) of 40,000 hours (Supplementary Fig. 3). This is the best stability performance among the recently published papers for the transfer-printed QD-LED devices (Supplementary Table 4).

Nevertheless, further improvements on the lifetime of our transfer-printed QD-LED device are still needed to lift the stability towards the level of spin-coated QD-LED devices. There are possible pathways to improve the stability of the QD-LED device. One is to optimise the core, shell and ligand materials of QD nanoparticles during synthesis to reduce the surface traps and to enhance the charge confinement effect²⁶. Another is to avoid exposing the QD layer from the air during the transfer-printing process since the oxygen or moisture in the air induces surface traps causing device degradation^{45,46}.

(line 823, page 35, Main article): References added

45. Chen, H.-C., Shabir, A., Tan, C. M., Singh, P. & Lin, J.-H. Degradation dynamics of quantum dots in white LED applications. *Sci. Rep.* **11**, 24153 (2021).
46. Moon, H., Lee, C., Lee, W., Kim, J. & Chae, H. Stability of quantum dots, quantum dot Films, and quantum dot light-emitting diodes for display applications. *Adv. Mater.* **31**, 1804294 (2019).

(page 4, Supplementary Information): Added

Supplementary Fig. 3: Measurement of device lifetime of QD-LED fabricated by the transfer printing process. a Luminance variation as a lapse of time for the red QD-LEDs at various initial luminances. **b** Extrapolation of the T_{50} at an initial luminance of 100 cd m^{-2} .

(page 18, Supplementary Information): Added

Supplementary Table 4: Summary of QD-LED device lifetimes reported from recent publications.

References	QD film process	QD Colour	Lifetime [hour]	Condition
Yang et al.^[17]	Spin coating	Red Green Blue	90,000 300,000 1,000	T_{50} at 100 cd m^{-2} T_{50} at 100 cd m^{-2} T_{50} at 100 cd m^{-2}
Dai et al.^[18]	Spin coating	Red	100,000	T_{50} at 100 cd m^{-2}
Pu et al.^[19]	Spin coating	Red Blue	90,000 10,000	T_{50} at 100 cd m^{-2} T_{50} at 100 cd m^{-2}
Weiran et al.^[20]	Spin coating	Red	2,200,000	T_{50} at 100 cd m^{-2}
Chen et al.^[21]	Spin coating	Red Blue	800 23	T_{90} at 1000 cd m^{-2} T_{50} at 1000 cd m^{-2}
Choi et al.^[10]	Transfer printing	Green	41.7 > 1900	T_{50} at 4554 cd m^{-2} T_{50} at 100 cd m^{-2}
Kim et al.^[13]	Transfer printing	Red	76.5 > 800	T_{50} at 1050 cd m^{-2} T_{50} at 100 cd m^{-2}
Kim et al.^[15]	Transfer printing	Red	35 > 700	T_{50} at 2037 cd m^{-2} T_{50} at 100 cd m^{-2}
This work	Transfer printing	Red	39,899	T_{50} at 100 cd m^{-2}

(page 20, Supplementary Information): Supplementary references added

- [10] Choi, M. K. *et al.* Wearable red–green–blue quantum dot light-emitting diode array using high-resolution intaglio transfer printing. *Nat. Commun.* **6**, 7149 (2015).
- [13] Kim, T.-H. *et al.* Full-colour quantum dot displays fabricated by transfer printing. *Nat. Photonics* **5**, 176–182 (2011).
- [15] Kim, B. H. *et al.* Multilayer transfer printing for pixelated, multicolor quantum dot light-emitting diodes. *ACS Nano* **10**, 4920–4925 (2016).
- [17] Yang, Y. *et al.* High-efficiency light-emitting devices based on quantum dots with tailored nanostructures. *Nat. Photonics* **9**, 259–266 (2015).
- [18] Dai, X. *et al.* Solution-processed, high-performance light-emitting diodes based on quantum dots. *Nature* **515**, 96–99 (2014).
- [19] Pu, C. *et al.* Electrochemically-stable ligands bridge the photoluminescence–electroluminescence gap of quantum dots. *Nat. Commun.* **11**, 937 (2020).
- [20] Cao, W. *et al.* Highly stable QLEDs with improved hole injection via quantum dot structure tailoring. *Nat. Commun.* **9**, 2608 (2018).
- [21] Chen, S. *et al.* On the degradation mechanisms of quantum-dot light-emitting diodes. *Nat. Commun.* **10**, 765 (2019).

Reviewer #2-(2) The authors are encouraged to put an explanation in the manuscript why the individual colors have different EQE even though it is obvious from the Figure, to give a clear understanding to the readers and make a better manuscript.

Response:

Thank you for the reviewer's valuable comment to encourage the authors to put an explanation in the manuscript for the reason why the individual QD-LEDs for each colour have different external quantum efficiencies (EQEs). The EQE of the QD-LED device is highly coupled with the material and device parameters, and the difference in EQE can be explained by the charge transport model. Based on the charge transport model, the EQE is mainly determined by the photoluminescence quantum yield (PLQY) of the QD layer, Langevin radiative recombination strength γ , Shockley-Read-Hall (SRH) recombination lifetime τ , and Auger capture probability C of the QDs (Eq. (12) in the manuscript). The EQE is also a function of the charge densities p^{QD} and n^{QD} of the hole and electron accumulated in the QD layer for a given applied voltage. Eq. (12) can be simplified to Eq. (R1) under the assumption of the charge-balanced condition ($p^{\text{QD}} = n^{\text{QD}}$)^{16,17}.

$$EQE = \eta_{\text{ext}}\eta_{\text{QD}} \frac{\gamma k}{1/(2\tau) + \gamma k + 2Ck^2} \quad (\text{R1})$$

Here, η_{ext} is an optical light extraction efficiency related to the multi-layered structure of the QD-LED device, and η_{QD} is the PLQY of the given QD layer. k denotes the charge carrier density at the centre of QDs. Under the assumption of charge-balanced condition, k equals p^{QD} or n^{QD} . Then, the maximum EQE of a given QD-LED device is determined by Eq. (R2) derived from $d(EQE)/dk = 0$.

$$EQE_{\text{max}} = \eta_{\text{ext}}\eta_{\text{QD}} \frac{Q}{2 + Q} \quad (\text{R2})$$

Here, a quality factor $Q = \gamma \times (\tau/C)^{1/2}$ is introduced to simplify the maximum EQE of the device. In Eq. (R2), the EQE_{max} is directly proportional to the PLQY η_{QD} and also is a function of the quality factor Q . Hence, the larger the quality factor and higher the PLQY of the QD-LED device, the higher EQE_{max} can be obtained.

First, the QD layer is considered as a close-packing of QD nanoparticles and is sandwiched by a hole transport layer (HTL) and an electron transport layer (ETL). Here, inter-particle fluorescence resonance energy transfer (FRET) within the QD layer or at the interface of the

QD layer can reduce the PLQY of the QD layer¹⁸. The inter-particle FRET is mainly determined by the surface configuration and the distance between two adjacent nanoparticles. Therefore, the PLQY can be considered as both the material- and device-level parameter since the properties of QDs directly affect PLQY itself and the interaction with the surrounding transport layers. Second, the quality factor is determined by the material-level parameters of QDs. Under the assumption of the identical Langevin radiative recombination strength, the quality factor depends on the SRH and Auger non-radiative recombination constants. The SRH recombination lifetime and the Auger capture probability are mainly related to the defective trap density and the carrier interaction with phonon¹⁹. Due to the different process conditions for synthesising different QDs, each QD can have different defective traps and phonon interactions in their nanocrystal cores.

The PLQY η_{QD} , SRH recombination lifetimes τ and Auger capture probabilities C and the quality factors Q extracted by the simulation and experiments (Fig. 3c) are listed in the revised Supplementary Table S3 for the respective red, green, cyan, and blue QD-LEDs. The maximum EQEs, PLQYs, quality factors, SRH and Auger recombination constants of each QD-LED are also plotted in Fig. R1 for graphical visualisation.

First, the red QD-LED shows the highest EQE_{max} and blue shows the lowest EQE_{max} , while the green and the cyan QD-LEDs show EQE_{max} in between the red and the blue QD-LEDs (Fig. R1a). The PLQYs of the QD layers are obtained to be 0.32, 0.04, 0.23 and 0.05 for the monochromatic red, green, cyan, and blue QD-LED devices (Fig. R1b). The PLQYs of the red and cyan QD layers show larger values than those of the green and blue QD layers. In terms of PLQY, we may guess that the red and cyan QDs have better layer and interfacial properties than green and blue QDs. Second, the quality factors are calculated to be 4.41, 5.94, 0.56 and 0.65 for the respective colour devices (Fig. R1c). The quality factors of the red and the green QD-LEDs are larger than those of the cyan and the blue QD-LEDs. Since the quality factor is determined by the SRH and Auger recombination constants, the higher the quality factor can be obtained with the longer SRH recombination lifetime and the smaller Auger capture probability. In Fig. R1d, the red and green QDs show longer SRH recombination lifetime and smaller Auger capture probability compared with the cyan and blue QDs, resulting in higher quality factors than those of cyan and blue QDs.

In summary, based on the charge transport model, the reason why the different EQEs are observed in the individual colour of the QD-LED device is that the QD layers have different properties in terms of both the quality factor and the PLQY of the QD layer. The quality factor

of the QD-LED device as a material-level parameter is determined by the recombination constants which are highly affected by the defective trap density and the photon interactions in the QD nanoparticle. The PLQY of the QD layer as a material- and device-level parameter is affected by inter-particle FRET within the QD layer or at the interfaces of the QD layer for the surrounding transport layers. A detailed study on the EQE performance for the QD material and device architecture is being undertaken by our group based on the charge transport simulation model and experiments. In accordance with the reviewer's comment, we have changed our manuscript as below to explain the reason for different EQEs of the individual colour of QD-LED.

Fig. R1: The relationship between the EQE and the other material parameters for different colours of QDs. a maximum EQE, **b** PLQYs of the QD layers, **c** quality factors of the QD layers, and **d** SRH recombination lifetime and Auger capture probability corresponding to the colour of QDs.

Corrections: -----

(line 299, page 13, Main article): Added

The fabricated red, green, cyan, and blue QD-LEDs show different maximum EQE due to their differences in material-level and device-level parameters affected by the QD materials and device architectures. Specifically, the reason why the different EQEs are observed in the individual colour of the QD-LED device is that the QD layers have different properties in terms

of both the photoluminescence quantum yield (PLQY) η_{QD} and the quality factor Q (Methods and Supplementary Table 3). The PLQY of QD layer as a material- and device-level parameter is affected by inter-particle fluorescence resonance energy transfer (FRET) within the QD layer or at the interfaces of the QD layer for the surrounding transport layers^{26,43}. The quality factor as a material-level parameter is determined by the recombination constants which are highly affected by the defective trap density and the photon interactions in the QD nanoparticle⁴⁴. Therefore, the differences in device EQE are attributed to the different material parameters and interfacial properties of QDs.

(line 819, page 35, Main article): References added

43. Roy, S. *et al.* Measurement of quantum yields of monolayer TMDs using dye-dispersed PMMA thin films. *Nanomaterials* **10**, 1032 (2020).
44. Piprek, J. Carrier Transport. in *Semiconductor Optoelectronic Devices* 49–82 (Elsevier, 2003).

(line 376, page 16, Main article): Modified

The red QD-LED shows the highest peak owing to its highest ~~photoluminescence quantum efficiency (PLQY)~~ of QD layer among the QDs (Supplementary Table 23).

(line 623, page 27, Main article): Added

Derivation of maximum EQE for QD-LED device

In Eq. (12), the EQE is mainly determined by the PLQY of QD layer, Langevin radiative recombination strength γ , SRH recombination lifetime τ , and Auger capture probability C of the QDs, and is a function of the charge densities p^{QD} and n^{QD} of the hole and electron accumulated in the QD layer for a given applied voltage. Eq. (12) can be simplified to Eq. (13) under the assumption of the charge-balanced condition ($p^{\text{QD}} = n^{\text{QD}}$)^{58,59}.

$$\underline{EQE = \eta_{\text{ext}}\eta_{\text{QD}} \frac{\gamma k}{1/(2\tau) + \gamma k + 2Ck^2}} \quad (13)$$

Here, η_{ext} is an optical light extraction efficiency related to the multi-layered structure of the QD-LED device, and η_{QD} is the PLQY of the given QD layer. k denotes the charge carrier

density at the centre of QDs, which is a function of the applied voltage. Under the assumption of charge-balanced condition, k equals p^{QD} or n^{QD} . Then, the maximum EQE of a given QD-LED device is determined by Eq. (14) derived from $d(\text{EQE})/dk = 0$.

$$\underline{\underline{EQE_{max} = \eta_{ext}\eta_{QD} \frac{Q}{2 + Q}} \quad (14)}$$

Here, a quality factor $Q = \gamma \times (\tau/C)^{1/2}$ is introduced, which is determined by the Langevin radiative recombination strengths γ , SRH recombination lifetime τ , and the Auger capture probability C . In Eq. (R2), the EQE_{max} is directly proportional to the PLQY η_{QD} and also is a function of the quality factor Q . Hence, the larger the quality factor and higher the PLQY of the QD-LED device, the higher EQE_{max} can be obtained.

(line 850, page 36, Main article): References added

58. Piprek, J. Efficiency droop in nitride-based light-emitting diodes: Efficiency droop in nitride-based light-emitting diodes. *Phys. Status Solidi A* **207**, 2217–2225 (2010).
59. Karpov, S. ABC-model for interpretation of internal quantum efficiency and its droop in III-nitride LEDs: a review. *Opt. Quantum Electron.* **47**, 1293–1303 (2015).

(page 17, Supplementary Information): Modified

Supplementary Table 23: Material parameters for red, green, cyan, and blue QDs used in the simulation.

Parameters	QDs			
	Red (CdSe/ZnS)	Green (CdSe/ZnS)	Cyan (CdSe/ZnS)	Blue (CdSe/ZnS)
ϵ_r^{QD}	9.4 ^[106]	9.4 ^[106]	9.4 ^[106]	9.4 ^[106]
Peak Wavelength [nm]	616	559	502	450
FWHM	20	20	20	20
Diameter [nm]	15.2	8.4	7.5	6.9
Number of QD layers	2	2	2	2
LUMO [eV]	-3.56	-3.45	-3.33	-3.19
HOMO [eV]	-5.54	-5.65	-5.77	-5.91
Optical Bandgap [eV]	1.98	2.20	2.44	2.72
μ_p^{QD} [cm ² V ⁻¹ s ⁻¹]	2.6×10 ⁻⁵	2.6×10 ⁻⁵	2.6×10 ⁻⁵	2.6×10 ⁻⁵
μ_n^{QD} [cm ² V ⁻¹ s ⁻¹]	2.6×10 ⁻⁵	2.6×10 ⁻⁵	2.6×10 ⁻⁵	2.6×10 ⁻⁵
σ_p	0.13	0.018	0.0085	0.009
σ_n	0.13	0.018	0.0085	0.009
η_{QD}	0.32	0.04	0.23	0.05
γ [cm ³ s ⁻¹]	1.0×10 ⁻¹¹	1.0×10 ⁻¹¹	1.0×10 ⁻¹¹	1.0×10 ⁻¹¹
τ [μs]	2.24	1.20	0.136	0.133
C [cm ⁶ s ⁻¹]	1.15×10 ⁻²⁹	0.34×10 ⁻²⁹	4.3×10 ⁻²⁹	3.16×10 ⁻²⁹
Q	4.41	5.94	0.56	0.65

Reviewer #2-(3) *Since the performance of the white light is deteriorated by low EQE colors, so what optimization can be done to improve the EQE for those colors.*

Response:

Thank you for the reviewer's valuable comments on the optimisation for improving the low device EQEs. As the reviewer commented, the performance of the pixelated QD-LED white lighting system can be deteriorated by the low EQE colour QDs. According to our investigation, the EQEs of our transfer-printed QD-LED devices fabricated in this study are similar to the state-of-the-art records (Supplementary Table 2 in revised Supplementary Information). However, to further enhance the EQEs, material-, device-, process-, and system-level design and optimisation of the white lighting system can be suggested as follows.

(i) Material-level design and optimisation of QD nanoparticle

As mentioned in the response to the reviewer #2's comment (2), the EQE of the device is related to the recombination constants and the PLQY of the QD materials. The recombination constants are determined by the defective trap and phonon interaction in the core of the QDs, and the PLQY depends on the inter-particle FRET among the QDs determined by the particle size and the surface configuration of QDs. One way to enhance the device EQE is to optimise QD synthesis process for minimising the defective traps by lattice mismatches between QD core/shell. Another way is to design and optimise the shell and surface-binding ligands. Many reports show that a thicker shell with short surface ligands is the best combination for the surface configuration to get higher electroluminescence (EL) performance²⁰. The QDs with thicker shells reduce FRET between QDs and increase the quantum confinement effect of QDs, and shorter surface ligands facilitate charge injection from the hole/electron transport layer to the QDs layers. The reduced FRET, enhanced quantum confinement effect and charge injection of the QD material results in the improvement of the device EQE. Therefore, by the optimising the surface configuration and materials of the QD nanoparticles, higher EQE of the device can be achieved.

(ii) Device-level optimisation of charge transport layer

In general, unbalanced charge injection causes a low device EQE and it is crucial to make balance in the charge injection of hole and electron by tailoring electrical properties of charge transport layers. Many reports have made efforts to employ various materials, thickness, and

surface treatment of transport layers for improving the charge balance^{21,22}. One approach to charge balance is to customise the charge transport layer for the individual QDs by tailoring material and layer properties such as energy-band level, carrier mobility, and thickness. Another approach is to use an interlayer between the QD layer and the electron transport layer. The interlayer can play as a barrier layer to reduce excessive electron injection into the QD layer, resulting in balanced charge injection for enhancement of device EQE¹¹. Besides, the interlayer serves as a lifting layer to enhance the pick-up yield and uniformity during the pick-up process in the transfer printing process²³. In summary, by customising the charge transport layers and by using the interlayer, the EQE of the individual QD-LED device can be enhanced in terms of device-level optimisation.

(iii) Process-level innovation by multilayer transfer printing

According to the device-level optimisation, the customisation of the charge transport layers for the individual QDs can be a promising approach to enhancing the EQE of each device. However, the transfer printing method used in this study allows the same ETL and HTL for the different QDs, since only the QD layers were transferred onto the same HTL and the ETL was spin-coated after the transfer printing of QD patterns. With this transfer printing process, each of the transport layers cannot be customised for the individual QD to enhance the EQE of each device. Multilayer transfer printing is an approach that transfer the QD layer together with the customised transport layers¹⁵. By using the multilayer transfer printing method, it is possible to pixelate the entire layer of QD-LED devices except the anode and cathode electrodes. Since the charge transport layers of individual QD-LED pixels are optimised to the individual QDs, the enhanced device EQE of each QD-LED can be expected, and the EQE of the pixelated QD-LED based white lighting system can be further improved.

(iv) System-level driving with independent anode electrodes

Once the EQE of each QD-LED device is maximised by the aforementioned methods, the system-level approach to enhance the EQE of the white lighting system is to drive each QD-LED pattern independently with the electrodes separately pixelated onto the each QD patterns. Normally, QD-LED exhibits maximum EQE at a specific voltage due to the combinational effect of radiative and non-radiative recombination processes, as described in the author's response to reviewer #2's comment (2). Therefore, the EQE of the white lighting system can be further enhanced, if we utilise totally patterned QD-LED devices at the voltage operating in

the highest EQE. An independent subpixel driving with pixelated electrodes enables the devices to operate at different voltages for their maximum EQEs. Here, the layout of emission patterns should be re-optimised in order to adjust the emission contributions of each QD-LED pattern for the targeted white chromaticity. This concept can be applied to high-end white lighting systems for enhanced efficiency with full-colour illumination, with the extra process and driving costs.

In conclusion, the white lighting system can be achieved by the aforementioned (i) material-level design of QD nanoparticles, (ii) device-level optimisation of charge transport layers, (iii) process-level innovation of transfer printing method, and (iv) system-level driving with independent anode electrodes. To implement a higher performance smart white lighting system, all approaches described above are currently being performed by our group. To address the reviewer's comment, the authors added and corrected the manuscript as follows to describe the methods for improving the EQE of QD-LED devices.

Corrections: -----

(line 276, page 12, Main article): Modified

To predict precisely the electro-optical characteristics of QD-LEDs by the device-level charge transport simulation, the material-level parameters should be extracted in advance by fabricating monochromatic QD-LED devices via transfer printing technique with the red, green, cyan, and blue QDs selected in the previous section.

(line 291, page 12, Main article): Added

In terms of device EQE, even though the transfer-printed QD-LED devices show lower EQE than those of the spin-coated devices, they are comparable to the state-of-the-art transfer-printed devices (Supplementary Table 2). The low EQEs of the individual devices can affect the performance of the white lighting system. Therefore, it is required to enhance the EQE of the transfer-printed QD-LED devices by material-level optimisation of QD nanoparticles and device-level customisation of transport layers^{38,39,22}. Furthermore, by the process-level innovation of the transfer printing technique and the system-level independent pixel driving method, the white lighting system with enhanced EL performance can be expected⁴⁰⁻⁴².

(line 809, page 35, Main article): References added

38. Kim, T. *et al.* Efficient and stable blue quantum dot light-emitting diode. *Nature* **586**, 385–389 (2020).
39. Dai, X. *et al.* Solution-processed, high-performance light-emitting diodes based on quantum dots. *Nature* **515**, 96–99 (2014).
40. Cho, K.-S. *et al.* High-performance crosslinked colloidal quantum-dot light-emitting diodes. *Nat. Photonics* **3**, 341–345 (2009).
41. Chiba, T. *et al.* Anion-exchange red perovskite quantum dots with ammonium iodine salts for highly efficient light-emitting devices. *Nat. Photonics* **12**, 681–687 (2018).
42. Kim, B. H. *et al.* Multilayer transfer printing for pixelated, multicolor quantum dot light-emitting diodes. *ACS Nano* **10**, 4920–4925 (2016).

Reviewer #2-(4) *The authors are encouraged to put an explanation about the change of color chromaticity with applied voltage.*

Response:

Thank you for the valuable comments to encourage the authors to put an explanation about the change of colour chromaticity with applied voltage. Regarding the reviewer's comment, we tried to explain the change of colour chromaticity for the applied voltage with the variation of the peak radiances at the peak wavelengths of the red, green, cyan, and blue QDs. The main reason for the change of colour chromaticity with applied voltage can be explained by the different voltage dependencies of the emission peak curves of each patterned QD-LED.

First, the threshold voltages of the fabricated red, green, cyan, and blue QD-LEDs used in this study are observed to be 2.0 V, 2.2 V, 2.6 V, and 3.0 V, respectively, owing to their different band-offset between the QD layers and the transport layers (HTL and ETL). Different QDs have different optical bandgaps according to their peak wavelengths and the optical bandgap determines the band-offset between the QD layer and the transport layers²⁴. In the Supplementary Fig. 5, the band-offsets of the red, green, cyan, and blue QD-LEDs are obtained to be 0.26 eV, 0.36 eV, 0.48 eV, and 0.63 eV, respectively. Since the hole and electrons are started to be injected after a certain threshold voltage overcoming the band-offset, the threshold voltages of the QD-LEDs become larger in the order of red, green, cyan, and blue QD-LEDs. Therefore, owing to the order of threshold voltages, the red, green, cyan, and blue QD-LED pixels are sequentially turned on as the voltage increases.

Next, the slopes of the emission peak curves with respect to the applied voltage are changed due to the new layout design optimised by our computational design framework. This can be clearly explained by the voltage variation of the simulated peak radiances at each peak wavelength of the individual QD-LED pattern. The variations of the peak radiances with respect to the applied voltage are plotted in Fig. R2. Figure R2 shows the simulated variation of the emission contributions at given peak wavelengths of each patterned QD-LED with respect to the applied voltage for the lighting systems with identical and optimised emission widths. In the lighting system with identical emission widths, since the red emission dominates other colours over the entire voltage range (Fig. R2a), the colour chromaticity is barely changed its colour from red.

Meanwhile, the slopes of the curves for emission peaks are changed by the newly designed pattern sizes in the white lighting system with optimised emission widths as shown in Fig. R2b.

In this study, to match the white chromaticity to D65 illuminant at a given voltage (at 5 V), we optimised the emission areas of each QD-LED pattern so that the emission areas of QD patterns are changed (for example, the red QD pattern became shrunk and the blue QD pattern became enlarged). Resultantly, the slope of the red emission curve became gentle while the slopes of green, cyan, and blue emission curves became steep after the optimisation of the emission area. Coupled with the different threshold voltage, since the dominant colour was changed from red to blue through green and cyan after a certain voltage as shown in Fig. R2b, the colour chromaticity of the lighting with optimised emission widths is widely changed from red to blue through white as shown in Fig. 4i.

To address the reviewer’s comment, simulated peak radiances of the lighting systems at each peak wavelength are additionally plotted as a function of the applied voltage as Supplementary Fig. 10 in the revised Supplementary Information. We also put the explanation of the colour chromaticity change with applied voltage in the manuscript. All the changes or corrections regarding the reviewer’s comment are described as follows.

Fig. R2: Emission contribution at peak wavelengths of each QD pattern on the lighting systems with respect to the applied voltage. Simulated peak radiance obtained at peak wavelengths of 616 nm (red), 559 nm (green), 502 nm (cyan), and 450 nm (blue) QD patterns for the comparison of the simulated lighting systems having **a** identical emission widths and **b** optimised emission widths.

Corrections: -----
(line 440, page 19, Main Article): Added

The changes of the colour chromaticity are explained by the variation of the peak radiances for each QD with respect to the applied voltage. The theoretical peak radiances obtained at peak wavelengths of red (616 nm), green (559 nm), cyan (502 nm), and blue (450 nm) QDs

with respect to the applied voltage is plotted in Supplementary Fig. 10 for the simulated lighting systems having identical and optimised emission weighting factors. Since each of the QD patterns is turned on in the order of red, green, cyan, and blue due to their different threshold voltages, the colour chromaticity can be changed by the applied voltage. In the reference lighting system with identical emission widths, the red emission always dominates other colours over the entire voltage range (Supplementary Fig. 10a), and the colour chromaticity of the reference lighting system is barely changed its colour from red. Meanwhile, the optimised white lighting system shows a wide chromaticity change with the voltage due to the change of voltage dependencies of emission curves caused by the new layout design of the pixelated QD patterns (Supplementary Fig. 10b).

(line 463, page 19, Main Article): Modified

In Fig. 41, the fabricated white lighting system with optimised emission widths also shows very wide colour changes from red to blue through white as ~~expected~~ explained by our computational design framework. Meanwhile, the fabricated lighting system with identical emission widths hardly changes its colour from red over the voltage due to the largest emission contribution of the red QD pattern.

(page 11, Supplementary Information): Added

Supplementary Fig. 10: Emission contribution at peak wavelengths of each QD pattern on the lighting systems with respect to the applied voltage. Simulated peak radiance at peak wavelengths of 616 nm (red), 559 nm (green), 502 nm (cyan), and 450 nm (blue) QD patterns for the comparison of the simulated lighting systems having **a** identical emission widths and **b** optimised emission widths.

Reviewer #3

Samarakoon et al. proposed a computational design framework for the first time to design the architecture of a white lighting system having a layout of multiple pixelated patterns of electric-field-driven quantum dot light-emitting diodes (QD-LEDs). The fabricated white lighting system using the transfer printing technique exhibits excellent lighting performances of 92% CRI and wide color temperature variation from 1612 K to 8903 K with only four pixelated QD-LEDs. This strategy of computational design can be targeted to direct the fabrication of the efficient and pixelated white QD-LEDs, and the work is novel. However, I think there still exists a few of problems:

Response:

Thank you sincerely for taking the time and effort in evaluating our manuscript and for the positive remarks with valuable comments. As we understand, reviewer #3' comments are related to the expandability of our computational design framework, EL performances of the optimised white lighting system, new design idea with identical emission widths, the reason for selecting the horizontal pitch of 375 μm , and discrepancy in requirements between our work and standard smart lightings. We think these are very insightful comments for upgrading our manuscript. We tried our best to address all the comments and questions. We hope that you find the improvement in the revised manuscript in accordance with your valuable review and comments.

Reviewer #3-(1) *Is this computational design just suitable for the fabrication of pixelated white QD-LEDs. Based on this method, whether the computational design can be further developed as to white QD-LEDs via other fabricating technology. And what does need to modify in this method.*

Response:

The authors appreciate the reviewer's insightful comments on the expandability of our proposed computational design framework to further applications and modifications for the various types of lighting system architectures fabricated by other process technologies. Basically, the methodology proposed in this study is to optimize the emission spectrum of a white lighting system with multiple QDs by virtue of QD colour tunability. The number of QD colours, peak wavelengths of QDs, and emission contribution of each QD is designed by our combinatorial colour optimization process for the spectrum design. Furthermore, through the computational simulation of various electro-optical characteristics of individual QD devices by the charge transport model, the pattern layout is precisely designed to follow the optimised emission contribution of each QD pattern. **Therefore, we can definitely declare that our computational design framework proposed in this study is the most powerful method to optimise the colour combination and the emission contribution of any kind of white lighting system with multiple primary colours. In this regard, our computational design framework can be, of course, used not only for pixelated white QD-LED but also for other types of various white lighting system architectures with similar operational principles.**

To date, several technologies achieving the white lighting system with multiple primary coloured QD-LEDs has been reported. The system architectures of QD-LED based white lightings are classified into a patterned-type^{6,7}, a stacked-type^{23,25}, and a mixed-type^{26,27} whereby the emissive layers (EMLs) of the devices are formed by different configurations of QDs. In the case of patterned-type white lighting, the transfer printing technology is the most promising fabrication process among other processes to pixelate the emission patterns of each primary QDs (More details are described in the response to the first comment of Reviewer #2). Regardless of the fabrication processes, if the white lighting system has pixelated QD patterns, our computational design framework provides a solution of design guidelines for selecting and optimising the combination of QDs and their emission contribution achieving both the high colour rendering index (CRI) and precise white chromaticity.

In contrast, stacked-type QD-LED white lightings have the EML of a multi-layered structure that consists of multiple stacks of heterogeneous QD layers, while mixed-type QD-LED white lightings have the EML of a compounded layer in which R, G, and B QD nanoparticles are mixed and randomly located. Stacked-type is fabricated via a multi-step spin-coating process by stacking vertically the R, G and B QD layers, and mixed-type is fabricated via a single-step spin-coating process by simply mixing R, G, and B QDs in the solution. In these cases, a huge number of experiments are required to empirically obtain the optimal combination of QDs and the emission contributions. However, in the recent reports related to these architectures, the colour properties of lighting had not been theoretically or systematically designed but empirically optimised with limited experiments.

In these types of white lightings, the stacked-type and the mixed-type QD-LED based white lighting system can be systematically designed by our computational design framework with respect to the design parameters for their specific system architectures. Here, the computational charge transport model needs to be modified according to the specific structure of white lighting system. Especially, the governing equation for the dynamic motion of the charge carriers should be changed for both the stacked- and mixed-type QD-LED white lighting. In the case of the stacked-type QD-LED white lighting system, the additional charge transport model describing the electric-field dependent carrier hopping process among the stacked QD layers should be included to model the energy-level difference in the heterogeneous stack of QD layers. In the mixed-type QD-LED white lighting, the model of the charge carrier decay caused by the inter-particle FRET and/or optical quenching/absorption among the QDs in the random QD mixture should be further developed and included.

Our computational charge transport model is even very useful to the full-colour QD-LED displays based on an active-matrix backplane where each pixel is driven by thin-film transistors. In the case of the full-colour EL-driven QD-LED display applications, triple or quadruple pixel structures can be employed to enhance the colour properties with three or four primary colours. By virtue of the QD colour (peak wavelength) tunability, it is expected that more vivid colour of the display system can be achieved. However, in the white balance optimisation of the display system with the triple or quadruple coloured pixel structure, the combination of QDs can be designed systematically by our combinatorial colour optimisation process with modification of the cost function for the targeted colour temperature. The emission contributions of individual subpixels for a given voltage can also be predicted by the luminance-voltage curves simulated by our charge transport model.

In summary, our computational framework is a total solution providing the design guideline for the optimal spectrum and predicting the electro-optical characteristics of the various QD-LED white lighting system architectures such as patterned-, stacked-, and mixed-types fabricated by different fabrication processes. According to the different target performances, the lighting or display system can be optimised by modifying the cost function in the combinatorial colour optimisation process or by additionally modelling the physical phenomena in the charge transport simulation model. In accordance with the reviewer's comment, we added a paragraph in the revised manuscript for the possible application fields of our computational design framework below.

Corrections: -----

(line 182, page 8, Main article): Added

Our computational framework can also be applied to the various types of system architectures such as patterned^{21,24}, stacked^{22,23}, and mixed^{15,37} types of QD-LEDs fabricated by different process techniques for various lighting systems. The spectrum of any kind of the QD-LED white lighting system based on multiple primary colours can be systematically designed by our combinatorial colour optimisation process. The charge transport simulation can also be applied to the various types of lighting system architecture with the simulation models modified for their QD configurations. Our computational design framework can be further used for the system design of the EL-based full-colour QD-LED displays with multiple primary colours. The colour optimisation process can be used for optimising the white balance, and the charge transport model can be used to predict the EL characteristics of each QD-LED subpixel with respect to the applied voltage. In summary, our computational design framework can be used for various smart lighting and display applications with similar operational principle.

(line 806, page 35, Main article): References added

37. Anikeeva, P. O., Halpert, J. E., Bawendi, M. G. & Bulović, V. Electroluminescence from a mixed red–green–blue colloidal quantum dot monolayer. *Nano Lett.* **7**, 2196–2200 (2007).

Reviewer #3-(2) Could the EL performance of white QD-LEDs in this work be provided? Where does this work stand in, compared with other white devices reported by literatures?

Response:

We appreciate your valuable comments and suggestions on the comparison of our EL performances with recent publications. As the reviewer commented, the authors provided our experimental EL performances such as EQE and current efficiency (CE) of the fabricated QD-LED lighting systems with identical and optimised emission widths as shown in Fig. R3. In Fig. R3, the fabricated white lighting system with the optimised emission widths shows maximally 1.4% of EQE, and the CE of 2.5 cd A^{-1} , while the lighting system with identical emission widths exhibits maximum EQE of 3.0% and the CE of 6.7 cd A^{-1} . The fabricated white lighting system with optimised emission shows lower EQE and CE than those of the white lighting system with identical emission widths, due to the shrinkage of the red QD-LED emission area having the largest EQE.

Fig. R3: Experimental EL performances of the white QD-LED lighting systems having the optimised (red) and identical (black) emission widths. A EQE and b CE of the white lighting systems with respect to the applied voltage.

We also provided the achievements on white and monochromatic QD-LEDs reported by recent publications to compare our results with others and to show where this work stands in. Here, the luminance, EQE, CE, CIE, CRI, and correlated colour temperature (CCT) of the representative monochromatic QD-LEDs and white QD-LED lighting systems are selected and listed in Table R2 for comparison. We listed EL performances of the monochromatic red, green,

and blue transfer-printed QD-LED devices and the white lighting systems from the publications. Our experimental results of the fabricated monochromatic QD-LEDs and the white QD-LED lighting system are also shown in the table for comparison. Our monochromatic QD-LEDs fabricated by the transfer printing technique shows the maximum EQEs of 4.6%, 0.6%, 1.0%, and 0.3% for red, green, cyan, and blue, respectively, with the luminances of 19,670 cd m⁻², 10,370 cd m⁻², 8070 cd m⁻², and 442 cd m⁻². The EL performances achieved in this study are similar levels to the recent publications for transfer printing techniques. Our white QD-LED lighting system also shows an EQE of 1.4% with a luminance of 3285 cd m⁻² which are similar to the results from the previous publications, and the CE of our white lighting system exhibits 2.5 cd A⁻¹ which is the best among the other types of QD-LED based white lighting systems.

Table R2. Performance summary of previous studies on QD-LED based white lighting systems. R, G, C, B, and W denotes the red, green, cyan, blue, and white, respectively.

References	Device features	Luminance [cd/m ²]	EQE [%]	Current Efficiency [cd/A]	CIE _{xy} (@ white)	CRI [%]	CCT [K]
	CIE D65 standard illuminant	-	-	-	(0.31, 0.33)	100	6500
This work	Transfer printed white QD-LEDs with RGB pixelation	19,670 (R)	4.3 (R)				
		10,370 (G)	0.6 (G)				
		8,070 (C)	1.0 (C)	2.5 (W)	(0.33, 0.32)	92	1612 – 8903
		442 (B)	0.3 (B)				
		3,285 (W)	1.4 (W)				
Choi et al. ⁷	Transfer printed white QD-LEDs with RGB patterning	4000 (W)	1.6 (W)	-	(0.39, 0.38)	-	-
Kim et al. ²³	Transfer printed white QD-LEDs with RGB stacking	3380 (W)	-	0.4 (W)	(0.36, 0.37)	-	-
Bae et al. ²⁶	Spin coated white QD-LEDs with RGB mixing	3220 (W)	0.9 (W)	-	(0.31, 0.32)	92	6874
Kim et al. ⁶	Transfer printed monochromatic QD-LEDs with RGB patterning	16,380 (R) 6,425 (G) 423 (B)	-	4 (R) 0.5 (G) 0.04 (B)	-	-	-
Nam et al. ²⁸	Transfer printed monochromatic QD-LEDs with thermodynamic route	14,063 (G)	3.3 (G)	14.8 (G)	-	-	-
Kim et al. ¹⁵	Transfer printed monochromatic QD-LEDs with multi-layer pick-up	1,200 (R) 600 (G) 30 (B)	2.3 (R) 1.4 (G) 0.03 (B)	-	-	-	-

In terms of the colour properties, our fabricated white lighting system also shows the CRI of 92% and the CIE_{xy} of (0.33, 0.32) which are comparable to the results from Bae *et al.*²⁶.

However, owing to the architectural uniqueness of our optimised QD-LED based white lighting, the colour of our white lighting system widely changes from red to blue with the wide variation of CCT from 1612 K to 8903 K, while the mixed type QD-LED exhibits the fixed colour with the CCT of 6874 K. Moreover, our white lighting system also exhibits a higher EQE than the mixed-type QD-LED white lighting device reported by Bae *et al.* From the comparison, our white lighting system optimised by the computational design framework exhibits the best performances in both of the EL and colour performances, compared with the other reports.

To address the reviewer’s comment, the authors provided Fig. R3 in the revised Supplementary Information as Supplementary Fig. 11 for the EL performances of our fabricated QD-LED white lighting system. The authors also added Table R2 in the revised Supplementary Information as Supplementary Table 2 to compare the EL performances of our optimised lighting system with the other publications. The corrections in the manuscript and Supplementary Information are listed below.

Corrections: -----

(line 458, page 19, Main article): Added

Our white QD-LED lighting system also shows the EL performances of 1.4% EQE and 2.4 cd A⁻¹ current efficiency which are comparable to the values from the previous reports on the various white QD-LED lightings (Supplementary Fig. 11 and Supplementary Table 2).

(page 12, Supplementary Information): Added

Supplementary Fig. 11: Experimental EL performances of the white QD-LED lighting systems having the optimised (red) and identical (black) emission widths. a EQE and b CE of the white lighting systems with respect to the applied voltage.

(page 16, Supplementary Information): Added

Supplementary Table 2: Performance summary of previous studies on QD-LED based white lighting systems. R, G, C, B, and W denotes the red, green, cyan, blue, and white, respectively.

References	Device features	Luminance [cd/m²]	EQE [%]	Current Efficiency [cd/A]	CIE_{xy} (@ white)	CRI [%]	CCT [K]
	CIE D65 standard illuminant	=	=	=	(0.31, 0.33)	100	6500
This work	Transfer printed white QD-LEDs with RGB pixelation	19,670 (R) 10,370 (G) 8,070 (C) 442 (B) 3,285 (W)	4.3 (R) 0.6 (G) 1.0 (C) 0.3 (B) 1.4 (W)	2.5 (W)	(0.33, 0.32)	92	1612 – 8903
Choi et al.^[10]	Transfer printed white QD-LEDs with RGB patterning	4000 (W)	1.6 (W)	=	(0.39, 0.38)	=	=
Kim et al.^[11]	Transfer printed white QD-LEDs with RGB stacking	3380 (W)	=	0.4 (W)	(0.36, 0.37)	=	=
Bae et al.^[12]	Spin coated white QD-LEDs with RGB mixing	3220 (W)	0.9 (W)	=	(0.31, 0.32)	92	6874
Kim et al.^[13]	Transfer printed monochromatic QD-LEDs with RGB patterning	16,380 (R) 6,425 (G) 423 (B)	=	4 (R) 0.5 (G) 0.04 (B)	=	=	=
Nam et al.^[14]	Transfer printed monochromatic QD-LEDs with thermodynamic route	14,063 (G)	3.3 (G)	14.8 (G)	=	=	=
Kim et al.^[15]	Transfer printed monochromatic QD-LEDs with multi-layer pick-up	1,200 (R) 600 (G) 30 (B)	2.3 (R) 1.4 (G) 0.03 (B)	=	=	=	=

(page 20, Supplementary Information): Supplementary references added

[10] Choi, M. K. et al. Wearable red–green–blue quantum dot light-emitting diode array using high-resolution intaglio transfer printing. *Nat. Commun.* **6**, 7149 (2015).

[11] Kim, T.-H. et al. Heterogeneous stacking of nanodot monolayers by dry pick-and-place transfer and its applications in quantum dot light-emitting diodes. *Nat. Commun.* **4**, 2637 (2013).

- [12] Bae, W. K. *et al.* R/G/B/Natural white light thin colloidal quantum dot-based light-emitting devices. *Adv. Mater.* **26**, 6387–6393 (2014).
- [13] Kim, T.-H. *et al.* Full-colour quantum dot displays fabricated by transfer printing. *Nat. Photonics* **5**, 176–182 (2011).
- [14] Nam, T. W. *et al.* Thermodynamic-driven polychromatic quantum dot patterning for light-emitting diodes beyond eye-limiting resolution. *Nat. Commun.* **11**, 3040 (2020).
- [15] Kim, B. H. *et al.* Multilayer transfer printing for pixelated, multicolor quantum dot light-emitting diodes. *ACS Nano* **10**, 4920–4925 (2016).

Reviewer #3-(3) Taking 4-primary colors for an example, as shown in Figure 4e, how to fabricate the high-quality white QD-LEDs with the identical emission widths? Namely, with the identical emission widths of 93.75 nm, which requirements do the corresponding QD materials need to satisfy based on the computational design method, and to realize for smart lightings. If the smart lightings can be realized based on the identical emission widths, what's the difference between white QD-LEDs with the identical emission widths and that with the optimised emission widths?

Response:

Thank you for the reviewer's valuable idea to design the white lighting system in a different way from our approach. It is thought to be a pretty good idea to design the white lighting system having identical emission widths with new QD materials. In order to examine the new design concept, we calculated theoretically system EQEs as EL performances of two white lighting systems having (i) the optimised emission widths with original QDs (this study, Case I) and (ii) the identical emission widths with modified QDs (reviewer's suggestion, Case II).

Case I. optimised emission widths with the given QD materials.

Case II. identical emission widths with optimal radiative recombination rates of modified QDs.

Here, Case I is the design of the white lighting system optimised originally in this study, and Case II is the design concept suggested by the reviewer. In Case II, the radiative recombination rates was selected for the charge transport model to modify the QD properties determining the EQEs of the devices as well as the white lighting system. As a key EL performance of the white lighting system, we defined a theoretical system EQE in Eq. (R3).

$$\text{System EQE [\%]} = 100 \times \eta_{ext} \sum_n (\eta_n a_n R_{RAD,n}) / \sum_n (a_n R_{TOT,n}) \quad (\text{R3})$$

Here, η_{ext} is a light extraction efficiency which is assumed to be 0.2 in this study. η_n and a_n are the PLQY and the emission weighting factor of the n -th QD pattern. The subscript n is an element of a set { R, G, C, B } for the four primary coloured lighting system with red (R), green (G), cyan (C), and blue (B) QDs. $R_{RAD,n}$ is a Langevin radiative recombination rate per unit area, and $R_{TOT,n}$ is the total summation of the Langevin recombination rate $R_{RAD,n}$, SRH recombination rate $R_{SRH,n}$ and Auger recombination rate $R_{AUG,n}$ for n -th colour QD. Prior to the

calculation of theoretical system EQEs, the R_{RAD} , R_{SRH} , and R_{AUG} for each of the monochromatic red, green, cyan, and blue QD-LED were obtained from our charge transport simulation under the white operation of 5 V and listed in Table R3. The PLQYs η of the respective QD-LEDs were also listed in the table. Here, a_n 's are set to be 1.0 for all the individual monochromatic QD-LED devices.

Table R3. Preparation of the simulation parameters of each monochromatic red, green, cyan, and blue QD-LED operating at 5 V.

Monochromatic QD-LED	Red	Green	Cyan	Blue
λ_n [nm]	616	559	502	450
η_n	0.32	0.04	0.23	0.05
a_n	1.0	1.0	1.0	1.0
R_{RAD} [$\text{cm}^{-2} \text{s}^{-1}$]	5.77×10^{17}	6.57×10^{17}	1.01×10^{17}	7.68×10^{16}
R_{SRH} [$\text{cm}^{-2} \text{s}^{-1}$]	8.07×10^{16}	1.16×10^{17}	3.19×10^{17}	2.62×10^{17}
R_{AUG} [$\text{cm}^{-2} \text{s}^{-1}$]	2.12×10^{17}	1.05×10^{17}	1.00×10^{17}	5.35×10^{16}

Table R4 summarises the simulation results for the hypothetical white lighting systems. According to the emission spectrum designed by the colour optimisation process in Fig. 2e, the relative ratio of the adjusted peak radiances, $A_R^*:A_G^*:A_C^*:A_B^*$, should be 0.75:0.85:0.93:0.99 for the power distribution of D65 at 616 nm, 559 nm, 502 nm, and 450 nm (Fig. 2f). For the given ratio of the adjusted peak radiances, the colour performances of 97% CRI and 0.022 Δ_{xy} with 7738 K CCT at 5 V can be simultaneously achieved according to the colour optimisation process. Therefore, as long as the emission spectrum of any kind of white lighting system follows the designed spectral radiance, the identical colour performances can be achieved whether the emission widths are optimized for the given QD materials or the QD materials are optimised for the identical emission widths.

First, the simulation results of the white lighting system (Case I) having the optimised emission widths with the original QD materials is described in the first section of the table. This lighting system was designed by our computational design framework suggested in the manuscript. Here, the emission weighting factors were optimised to be 0.017, 0.118, 0.130 and 0.735 for red, green, cyan, and blue QD-LED patterns. As a result, the white lighting exhibits a CRI of 97% with a Δ_{xy} of 0.022 at 5 V. However, due to the largest blue QD pattern, the

system EQE of the white lighting system having the optimised emission widths is theoretically obtained to be 0.5% by Eq. (R3) for the given QD-LEDs having the original device EQEs of 4.2% (red), 0.6% (green), 0.9% (cyan), and 0.2% (blue).

Table R4. Recombination rates, relative radiances and system EQE of the pixelated white QD-LED lighting systems simulated for the different design conditions and QD materials.

Conditions	Parameters	Red	Green	Cyan	Blue
Case I. Optimised emission width with current QDs	a_n	0.017	0.118	0.130	0.736
	η_n	0.32	0.04	0.23	0.05
	$a_n \times R_{\text{RAD}} [\text{cm}^{-2} \text{s}^{-1}]$	9.57×10^{15}	7.76×10^{16}	1.31×10^{16}	5.65×10^{16}
	$a_n \times R_{\text{SRH}} [\text{cm}^{-2} \text{s}^{-1}]$	1.34×10^{15}	1.37×10^{16}	4.13×10^{16}	1.93×10^{17}
	$a_n \times R_{\text{AUG}} [\text{cm}^{-2} \text{s}^{-1}]$	3.52×10^{15}	1.24×10^{16}	1.3×10^{16}	3.94×10^{16}
	$A_n^* [\text{W m}^{-2} \text{sr}^{-1} \text{nm}^{-1}]$	0.029	0.033	0.036	0.038
	Relative ratio of A_n^*	0.751	0.852	0.926	0.993
	EQE [%]	4.2	0.6	0.9	0.2
	System EQE [%]		0.51		
Case II. Identical emission width with optimised QDs (radiative recombination rates)	a_n	0.25	0.25	0.25	0.25
	η_n	0.32	0.04	0.23	0.05
	$a_n \times R_{\text{RAD}} [\text{cm}^{-2} \text{s}^{-1}]$	1.44×10^{17}	1.17×10^{18}	1.98×10^{17}	8.5×10^{17}
	$a_n \times R_{\text{SRH}} [\text{cm}^{-2} \text{s}^{-1}]$	2.02×10^{16}	2.9×10^{16}	7.98×10^{16}	6.55×10^{16}
	$a_n \times R_{\text{AUG}} [\text{cm}^{-2} \text{s}^{-1}]$	5.30×10^{16}	2.63×10^{16}	2.5×10^{16}	1.34×10^{16}
	$A_n^* [\text{W m}^{-2} \text{sr}^{-1} \text{nm}^{-1}]$	0.438	0.492	0.539	0.577
	Relative ratio of A_n^*	0.751	0.852	0.926	0.993
	EQE [%]	4.2	0.8	3.0	0.9
	System EQE [%]		1.36		

Next, the simulation results of the white lighting system (Case II) having identical emission widths with the modified radiative recombination rates of new QD materials are described in the second section of the table. The emission weighting factors a_n 's are set to be 0.25 for the identical emission widths. The radiative recombination rates of green, cyan, and blue QD materials are changed to match the peak radiances to the given ratio of 0.75:0.85:0.93:0.99. To match the spectrum by enhancing the emission contribution of green, cyan, and blue QD-LED patterns, very large amounts of the radiative recombination rates are required, and their corresponding device EQEs should also be increased to 0.8%, 3.0% and 0.9% from the original EQEs of 0.6%, 0.9% and 0.2%. With the enhanced device EQEs of the respective QD-LEDs, the hypothetical white QD-LED can reach the theoretical system EQE of 1.36% which is almost three times higher than that of the originally designed white lighting system with different emission widths. Of course, since the emission spectrum is the same as the spectrum

designed by the colour optimisation process, the colour performances are equivalent to the original design in Case I.

Resultantly, it was possible to design the white lighting system with identical emission widths. With identical emission widths, the peak radiances were controlled by the radiative recombination rates. By optimising the radiative recombination rates of the new QD combination, the requirements of QDs for the white lighting with the identical emission widths were calculated in terms of the device EQEs. It was found from our calculation that the white lighting system having the identical emission widths with the optimised QDs could have enhanced system EQE compared with the original design in the manuscript. However, it is currently difficult to directly control the radiative recombination rate to enhance the emission contribution of the QDs, due to the complexity of controlling the QD material parameters. Nevertheless, if the synthesise technology that the material constants of QD can be easily controlled is further developed in the near future, the method conceived by the reviewer is thought to be another promising approach to enhance the system EQE of white lighting. To address the reviewer's comment, we added several sentences in the revised manuscript. We also corrected minor typographical errors for the optimised emission weighting factors and their corresponding emission widths which were found during progressing this analysis.

Corrections: -----

(line 135, page 6, Main article): Added

For the optimal emission spectrum, there are two ways to control the emission contribution of the QD patterns to the emission spectrum of the white lighting system satisfying high CRI and the chromaticity of CIE D65 illuminant. One is to optimise the QD materials for identical emission widths of QD patterns, and the other is to optimise the emission widths of QD patterns for the given QD materials. Since it is difficult to control the material parameters of QDs for the identical emission widths, the emission widths of the pixelated QD patterns are optimised for a given QD materials in this study.

(line 397, page 17, Main article): Corrected

From the design rule in Eq. (2), the weighting factors for the optimised lighting system is designed to be 0.01~~67~~, 0.118, 0.13~~70~~, and 0.7~~2935~~ for the red, green, cyan, and blue QD patterns to have the emission spectrum satisfying the maximum CRI and D65 chromaticity at the applied voltage of 5 V (Inset in Fig. 4c).

(line 419, page 18, Main article): Corrected

The optimal emission widths for red, green, cyan, and blue QD patterns are designed to be 6 μm , 44 μm , 5149 μm , and 2746 μm for the respective optimised weighting factors of 0.0167, 0.118, 0.1370, and 0.72935.

Reviewer #3-(4) Why is 375 μm selected as the horizontal pitch P of the QD pixel group consisting of four QD patterns? Whether the pixel size can be further reduced, maybe the higher the pixel, the more useful for pixelated QD-LEDs? The EL performance maybe related to the pixel size of devices. When the pixel size is reduced, how the EL performance of white QD-LEDs will be show? Whether this or similar computational design methodology can be targeted to direct the relation of pixel size to the device performance?

Response:

Thank you for the reviewer's valuable comments on the reason for selecting the 375 μm pixel pitch and the dependency of pixel size on the EL performance of the patterned QD-LED based white lighting system. This is another important research topic that should be studied in detail by further studies on the optimisation of the pattern size regarding human perception and EL performances. In this study, there are two main reasons for selecting the horizontal pitch P of the QD pixel group to be 375 μm in this study.

First, the lighting system was designed based on the human perception of the pixel at a given distance. The minimum pixel group pitch which cannot be perceived by the human eye is expressed by Eq. (R4) for the minimum angular resolution $\Delta\phi$ of the human visual system and the distance D of the observer from the lighting system.

$$P < 2D \cdot \tan(\Delta\phi/2) \quad (\text{R4})$$

Here, the minimum angular resolution $\Delta\phi$ for a visual acuity of 20/20 is $(1/60)^{\circ 29}$. The perception limit of the human eye with respect to the distance from the lighting system is plotted in Fig. R4. Assuming that the average distance of the lighting at home is around 1.5 m, it is observed from the figure that the pixel group pitch should be less than 400 μm in order to make the lighting pattern unperceived at the distance. In our design, the entire horizontal emission width of 3.0 mm was divided into 8-pixel groups with R/G/C/B QD patterns, hence the horizontal pixel group pitch was designed to be 375 μm . Based on the human perception, this pixel group pitch could no longer be recognized by the human eye having a visual acuity of 20/20 at a distance of more than 1.3 m from the lighting system.

Fig. R4: Perception limit of pixel group pitch with respect to the distance of lighting from the user. The red line shows the perception limitation of pixel pitch for the lighting distance, and the pixel group pitch below the limitation cannot be recognized by the user at the given distance.

Second, to examine the relationship between the pixel group pitch and the EL performances experimentally, QD-LED lighting systems with the pixel group pitches P of 375 μm and 1500 μm were fabricated and tested. Figure R5 shows the experimental EL performances of the fabricated lighting systems for the cases of $P = 375 \mu\text{m}$ and $P = 1500 \mu\text{m}$ with the insets of their EL snapshots. Here, QD-LED lighting systems with identical emission pattern widths have been used in the experiments, hence each of the subpixel pitches is equally set to be 93.75 μm and 375 μm for the pixel group pitches of 375 μm and 1500 μm , respectively (Insets in Fig. R5a). In Fig. R5a, the maximum EQE of the lighting systems are measured to be 3.0 % and 1.9 % for the cases of $P = 375 \mu\text{m}$ and $P = 1500 \mu\text{m}$, respectively, showing that the lighting with smaller pixel pitch exhibits higher EQE than the lighting with larger pixel pitch. Figure R5b shows the current-voltage-luminance curves for the lighting systems having different pixel group pitches. The lighting system with 375 μm pitch exhibits a luminance of 18,540 cd m^{-2} which is almost double the luminance of the lighting system with 1500 μm pitch. From the results, the lighting system with the 375 μm pixel pitch shows higher performances in EQE and brightness than the lighting system with the 1500 μm pixel pitch.

These results indicate that the EL performances could depend on the pixel pitch of the pixelated lighting system. It might be explained by the size-dependent film quality of QD patterns affected by pick-up and transfer steps during the transfer printing process or by the local field enhancement of edge effect caused by the finite size of the QD patterns. However, a further study on the pattern size dependency of the EL properties for the transfer-printed white QD-LED lighting system is still required to be investigated in detail.

The relationship between the pixel pattern size and the device performance has not been analysed by the theoretical simulation model in this study. The QD patterns have widths of more than tens of micrometres and their thicknesses are around tens of nanometres (aspect ratios of more than 10^3). In this case, since the device layout can be assumed as an infinite plane, the one-dimensional charge transport simulation model along the vertical direction of the device plane is enough to simulate the electro-optical properties of each QD-LED device. However, if the aspect ratio of the QD pattern is much smaller than our design, it is thought that the EL performances of the pixelated QD-LEDs may be highly affected by the edge effect of the fine QD pattern. In order to analyse the relationship between the QD pattern size and the EL performance theoretically, the edge field effect of the fine QD pattern can be simulated by two- or three-dimensional charge transport model in QD-LED devices with the geometrical/architectural consideration. However, as the reviewer already know, this theoretical study of multi-dimensional simulation is another big research topic to be further developed. In order to focus on the colour design process with the patterned QD-LEDs and its validation, the size-dependency of the EL properties was tried only by the experimental fabrication, not by multi-dimensional simulation in this letter.

Currently, our group is conducting research on various QD pattern sizes for the QD-LED display application through the transfer printing technique. In particular, The trends in the EL performances are being analysed for 5 – 10 μm pattern width for Metaverse VR/AR and mobile applications, 50 – 100 μm pattern width for monitor or TV applications, and 1 – 3 mm pattern width for digital signage applications. Based on this comprehensive research, it is expected that the overall trends on EL characteristics for the pattern width can be finally concluded in the near future. In summary, there were two main reasons for selecting the pixel pitch of 375 μm . The pixel pitch of 375 μm was mainly determined by considering the aspects of human perception and the dependency on the EL performance from experiments. Regarding the reviewer's question, the author presented Figs. R4 and R5 in the revised Supplementary Information as Supplementary Figs. 7 and 8, and added sentences in the revised manuscript as below, in order to explain the reason for selecting the 375 μm pixel pitch in this study.

Fig. R5: Experimental electro-optical properties of the reference lighting systems having identical subpixel pattern widths for the horizontal pitches P of 375 μm and 1500 μm . **a** EQE-current curves of the reference lighting systems for $P = 375 \mu\text{m}$ and 1500 μm . Insets are the EL snapshots of the respective fabricated reference lighting systems. **b** current-voltage-luminance (J - V - L) curves of the reference lighting system having identical subpixel pattern widths for the horizontal pitches of 375 μm and 1500 μm .

Corrections: -----

(line 410, page 17, Main article): Added

The horizontal pitch P was determined by considering two aspects of (i) the human perception of the pixel group and (ii) the dependency of the pixel pitch on the EL performance in our experiments. First, by selecting the horizontal pitch to be 375 μm , the pixel group was designed not to be recognised by the human visual system at a usual distance of the lighting environment (Supplementary Fig. 7). Second, from our experiment, we selected 375 μm horizontal pitch since the lighting system with 375 μm pitch showed higher EL performance compared with the lighting system with 1500 μm (Supplementary Fig. 8).

(page 8, Supplementary Information): Added

Supplementary Fig. 7: Perception limit of pixel group pitch with respect to the distance of lighting from the user. The red line shows the perception limitation of pixel pitch for the lighting distance, and the pixel group pitch below the limitation cannot be recognized by the user at the given distance.

(page 9, Supplementary Information): Added

Supplementary Fig. 8: Experimental electro-optical properties of the reference lighting systems having identical subpixel pattern widths for the horizontal pitches P of 375 μm and 1500 μm . **a** EQE-current curves of the reference lighting systems for $P = 375 \mu\text{m}$ and 1500 μm . Inset are the EL snapshots of the respective fabricated reference lighting systems. **b** current-voltage-luminance (J - V - L) curves of the reference lighting system having identical subpixel pattern widths for the horizontal pitches of 375 μm and 1500 μm .

Reviewer #3-(5) *Smart lightings are expected to respond to the personal mood or circadian rhythm in the daily ambient living while also rendering colours of objects more accurately. Except that, I want to know more detailed requirements for the smart lightings. What's more, whether the discrepancy is still significant between the results in this work and the standards for smart lightings?*

Response:

We appreciate the reviewer's inquiry into smart lighting standards as it gives us an opportunity to highlight the current state of the industry and how our work fits with it. "Smart lighting", as it is commonly used, is a term used to refer to lighting technologies that are user and application aware. Major companies such as Philips, Osram, Samsung, and LG have been studied smart lighting from the early stage of the research field. **It is quite a young field with wide-reaching application domains, and as such there are no explicit standards.** This is similar to the domain of "Internet of Things" where the term encompasses a wide range of technologies unified their ability to work together by communicating via the internet. However, similar to how each IoT system will be subject to domain specific standards, "smart lighting" systems inherit standards established for conventional lighting systems by the CIE, Illumination Engineering Society (IES), and the American National Standard Institute (ANSI)³⁰.

Most commercially available smart lights are "smart" by virtue of being able to control their colour and luminance remotely via an app or to a schedule set by the user [<https://www.philips-hue.com/en-gb>, <https://www.ikea.com/gb/en/cat/smart-lighting-36812/>]. In this work, we are specifically focused on creating a smart lighting system that is suitable for general illumination (i.e., for lighting living spaces and office environments) that is also capable of adapting changing its colour and rendering precisely surrounding objects for ambient assisted living in a cost-effective way. **In this case, high CRI and high colour controllability with cost-effectiveness are essential requirements for the application.** First, the designs presented in our work is different with the other lighting technologies in the way of cost-effectiveness in terms of driving scheme and the large area manufacturability. It uses passive-matrix driving which individual colour is controlled with the same voltage at a time by employing common electrodes for all the devices in the white lighting system. This reduces the driving cost of the device while still allowing the colour variation by simply varying the driving voltage. Also, it uses QDs for the light generation which allows the device to be fabricated using a solution

process suitable for large area manufacturing. The fabrication process coupled with passive-matrix driving allows our architecture to be used to create large electroluminescent surfaces at a lower manufacturing and driving cost than was previously possible.

In the case of colour rendering capability, we have leveraged the colour purity of the QDs to create a device with extremely high CRI which is essential for general lighting applications for both visual comforts and for accurate colour reproduction. Extremely high CRI of 92% achieved in this study makes our white lighting system suitable for use in a wide range of applications from residential and road illumination (which usually require CRI ~ 70%) to applications with more strict optical requirements like art galleries and medical applications (which usually demands CRI > 90%). For comparison, a typical LED white light from Phillips has a CRI of around 80% with no colour controllability [<https://www.lighting.philips.co.uk/consumer/p/functional-ceiling-light/318143116/specifications>] while a Phillips Hue smart light with colour controllability has a CRI in the range 85% - 91% [<https://medium.com/simplebulb/is-smart-lighting-healthy-903ccc81c42b>]. Especially, the CRI-R9 value is crucial for expressing red, especially for the accurate skin colour reproduction, which is vital in many industries from filmmaking and textiles to medical applications. Unfortunately, in the latter case of colour controllable Philips Hue smart light, the CRI-R9 value is around 30% - 40% and can go as low as 11%. In contrast, our design shows a high CRI-R9 of 91.8% which is a sufficient value for the accurate rendering of red colour (see Supplementary Fig. 9).

Regarding colour controllability, it is also very important to express the exact chromaticity expected from the specified colour temperature of the white lighting system. However, it is not enough to use only the CCT value since the colours of the lightings can be different even for the identically specified CCT values. The colour can be more exactly specified by employing the CCT combined with the distance from the Planckian locus on the CIE UV colour space Δ_{uv} . The smaller the distance from the Planckian locus, the more exact colour can be expressed for the specified CCT. Lighting devices with thin-film transistor-based active-matrix switching are able to access a wide colour gamut, and exactly matching the Planckian locus is trivial for such a system. The challenge is designing a system with simple low-cost passive-matrix driving that can still predictably change its CCT and with a chromaticity locus closely matching the Planckian locus. In Fig. R6, we have highlighted the colour accuracy of our optimised white lighting system using the distance from the Planckian locus Δ_{uv} . As shown in Fig. R6, the voltage-dependent colour locus of our optimised white lighting system closely follows the

Planckian locus with the range of $-0.013 < \Delta_{uv} < +0.007$. Here, the isothermperature lines perpendicular to the Planckian locus represent $\Delta_{uv} < \pm 0.05$ range. The locus of colour achievable by our device traces the Planckian locus with a maximum absolute deviation from the Planckian locus $|\Delta_{uv}|$ of 0.013 which is almost 4 times smaller than the isothermperature criteria of 0.05. Therefore, as the colour locus of our design traces closely to the Planckian locus, the white lighting system optimised in this study can express more precise colour chromaticity for the specified CCT at a given voltage.

It is well known that the user preference for lighting colour is affected by (i) the circadian rhythm for daily life (temporal difference), (ii) region/culture (spatial difference), or (iii) lighting context (situational difference)^{31,32}. For example, low CCT (“warm”) lighting has been found to be better for relaxation while light with high CCT (“cool”) light has been found to be better at increasing attentiveness^{32,33}. Furthermore, it is also known that the structure of the light influences the customers’ mood. Direct lighting has been found to increase positive moods while ambient lighting has been found to decrease negative moods better³⁴. This has been shown to have complementary effects when coupled with low CCT illumination showing an impact on calming users’ anxiety³⁵. As such a smart lighting system for general illumination should be able to adapt its optical properties to match these users’ needs. In this context, we believe that the white lighting system presented in this work can cover a wide range of form factors for a diverse range of requirements.

In summary, the high CRI, precise colour, and wide colour controllability coupled with the cost-effectiveness make the proposed architectures ideal for human-centric lighting applications. We also believe that our approach that tries to address the problem of smart lighting in a holistic, end-to-end way would motivate discourse into establishing industry standards for smart lighting, in the future. To address the reviewer’s comment, we have added a figure showing the locus traced by our optimised white lighting system relative to the Planckian locus as a function of the applied voltage. We have also added a figure showing the deviation of our device’s locus from the Planckian locus. During the revision, we found an error in the original Supplementary Fig. 6 and replaced it with revised Supplementary Fig. 9. Finally, we have revised our manuscript to elaborate on the colour performances expected from smart lighting devices.

Fig. R6: Experimental colour variation of the optimised QD-LED white lighting system. a Colour locus and **b** distance from Planckian locus Δ_{uv} with respect to the applied voltage.

Corrections: -----

(line 469, page 20, Main article): Added

Owing to the wide colour controllability, the optimised white lighting system is able to respond to most of the colour preferences affected by the circadian rhythm for daily life (temporal difference), region/culture (spatial difference), or lighting context (situational difference)^{48–52}. In addition, the colour locus of the optimized white lighting system traces closely to the Planckian locus over the entire voltage range, enabling it to express a more precise colour close to the specified CCT (Supplementary Fig. 12). Together with the high CRI and wide CCT variation, the colour accuracy of our optimised white lighting system for the specified CCT would motivate discourse toward establishing industry standards for smart lighting.

(line 478, page 20, Main article): Corrected

~~Therefore~~In summary, the architecture of the lighting system optimised in this study shows excellent lighting properties for next-generation versatile smart lighting, which provides accurate colour rendering while also allowing wide colour controllability with high colour precision.

(line 484, page 20, Main article): Corrected

~~In conclusion~~Therefore, our ~~novel~~ computational design framework with combinatorial colour

optimisation process using the Nelder-Mead algorithm and complete charge transport simulation using the electric-field dependent charge injection model is a very useful methodology for designing and predicting the electro-optical properties of multi-primary white lighting systems based on pixelated QD-LEDs patterned by the transfer printing technology.

(line 830, page 36, Main article): References added

48. Davis, K. *et al.* Effects of ambient lighting displays on peripheral activity awareness. *IEEE Access* **5**, 9318–9335 (2017).
49. Wang, Q., Xu, H., Zhang, F. & Wang, Z. Influence of color temperature on comfort and preference for LED indoor lighting. *Optik* **129**, 21–29 (2017).
50. Delay, E. R. & Richardson, M. A. Times estimation in humans: Effects of ambient illumination and sex. *Percept. Motor. Skill* **53**, 747–750 (1981).
51. Hsieh, M. Effects of illuminance distribution, color temperature and illuminance level on positive and negative moods. *J. Asian Archit. Build.* **14**, 709–716 (2015).
52. Kuijsters, A., Redi, J., de Ruyter, B. & Heynderickx, I. Lighting to make you feel better: Improving the mood of elderly people with affective ambiances. *PLOS ONE* **10**, e0132732 (2015).

(page 13, Supplementary Information): Added

Supplementary Fig. 12: Experimental colour variation of the optimised QD-LED white lighting system. a Colour locus and **b** distance from Planckian locus Δ_{uv} with respect to the applied voltage.

(page 10, Supplementary Information): Corrected

Supplementary Fig. 69: CRI Bar charts of the optimised 4-primary white lighting system from the simulation and device fabrication. a Theoretical CRI values obtained from the simulation. **b** Experimental CRI values obtained from the fabricated device.

Appendix A: Lifetime testing method

Lifetime test was carried out under accelerated conditions to shorten the testing period, as widely employed and accepted for display/lighting industry. The lifetimes of QD-LEDs at various luminance were measured from the luminance degradation curves with respect to time at a given initial luminance (Fig. R7a). A lifetime T_{50} at a given luminance L can be extrapolated based on the experimental lifetime T_{50}^* measured at the experimental luminance L^* (Fig. R7b).

$$T_{50} = T_{50}^* \times \left(\frac{L^*}{L}\right)^n \quad (\text{R5})$$

The acceleration factor n is obtained by sampling the device at certain stresses of luminance ranging from 10,000 to 30,000 cd m^{-2} , and the respective lifetimes were measured. These data were fitted to calculate the value of n . The n factor was calculated to be 1.584. From the results, T_{50} lifetime under 100 cd m^{-2} initial luminance for our red QD-LEDs were extrapolated to be 39,899 hours, as shown in Fig. R7b. The QD-LEDs were encapsulated with commercially available UV-curing epoxy and cover glass and the lifetime tests were performed under ambient conditions. The devices were driven by a source meter (Keithley 2400) coupled with photonic multichannel analyser PMA-12 (Hamamatsu Photonics K.K.).

Fig. R7: Measurement of device lifetime of QD-LED fabricated by the transfer printing process. a Luminance variation as a lapse of time for the red QD-LEDs at various initial luminances. **b** Extrapolation of the T_{50} at an initial luminance of 100 cd m^{-2} .

References

1. Bang, S. Y. *et al.* Technology progress on quantum dot light-emitting diodes for next-generation displays. *Nanoscale Horiz.* **6**, 68–77 (2021).
2. Yang, J. *et al.* High-resolution patterning of colloidal quantum dots via non-destructive, light-driven ligand crosslinking. *Nat. Commun.* **11**, 2874 (2020).
3. Shulga, A. G. *et al.* Patterned quantum dot photosensitive FETs for medium frequency optoelectronics. *Adv. Mater. Technol.* **4**, 1900054 (2019).
4. Jia, S. *et al.* High performance inkjet-printed quantum-dot light-emitting diodes with high operational stability. *Adv. Opt. Mater.* **9**, 2101069 (2021).
5. Deegan, R. D. *et al.* Capillary flow as the cause of ring stains from dried liquid drops. *Nature* **389**, 827–829 (1997).
6. Kim, T.-H. *et al.* Full-colour quantum dot displays fabricated by transfer printing. *Nat. Photonics* **5**, 176–182 (2011).
7. Choi, M. K. *et al.* Wearable red–green–blue quantum dot light-emitting diode array using high-resolution intaglio transfer printing. *Nat. Commun.* **6**, 7149 (2015).
8. Chen, H.-C., Shabir, A., Tan, C. M., Singh, P. & Lin, J.-H. Degradation dynamics of quantum dots in white LED applications. *Sci. Rep.* **11**, 24153 (2021).
9. Moon, H., Lee, C., Lee, W., Kim, J. & Chae, H. Stability of quantum dots, quantum dot Films, and quantum dot light-emitting diodes for display applications. *Adv. Mater.* **31**, 1804294 (2019).
10. Yang, Y. *et al.* High-efficiency light-emitting devices based on quantum dots with tailored nanostructures. *Nat. Photonics* **9**, 259–266 (2015).
11. Dai, X. *et al.* Solution-processed, high-performance light-emitting diodes based on quantum dots. *Nature* **515**, 96–99 (2014).
12. Pu, C. *et al.* Electrochemically-stable ligands bridge the photoluminescence-electroluminescence gap of quantum dots. *Nat. Commun.* **11**, 937 (2020).
13. Cao, W. *et al.* Highly stable QLEDs with improved hole injection via quantum dot structure tailoring. *Nat. Commun.* **9**, 2608 (2018).
14. Chen, S. *et al.* On the degradation mechanisms of quantum-dot light-emitting diodes. *Nat. Commun.* **10**, 765 (2019).
15. Kim, B. H. *et al.* Multilayer transfer printing for pixelated, multicolor quantum dot light-emitting diodes. *ACS Nano* **10**, 4920–4925 (2016).
16. Piprek, J. Efficiency droop in nitride-based light-emitting diodes: Efficiency droop in

- nitride-based light-emitting diodes. *Phys. Status Solidi A* **207**, 2217–2225 (2010).
17. Karpov, S. ABC-model for interpretation of internal quantum efficiency and its droop in III-nitride LEDs: a review. *Opt. Quantum Electron.* **47**, 1293–1303 (2015).
 18. Roy, S. *et al.* Measurement of quantum yields of monolayer TMDs using dye-dispersed PMMA thin films. *Nanomaterials* **10**, 1032 (2020).
 19. Piprek, J. Carrier Transport. in *Semiconductor Optoelectronic Devices* 49–82 (Elsevier, 2003).
 20. Kim, T. *et al.* Efficient and stable blue quantum dot light-emitting diode. *Nature* **586**, 385–389 (2020).
 21. Cho, K.-S. *et al.* High-performance crosslinked colloidal quantum-dot light-emitting diodes. *Nat. Photonics* **3**, 341–345 (2009).
 22. Chiba, T. *et al.* Anion-exchange red perovskite quantum dots with ammonium iodine salts for highly efficient light-emitting devices. *Nat. Photonics* **12**, 681–687 (2018).
 23. Kim, T.-H. *et al.* Heterogeneous stacking of nanodot monolayers by dry pick-and-place transfer and its applications in quantum dot light-emitting diodes. *Nat. Commun.* **4**, 2637 (2013).
 24. Jung, S.-M. *et al.* Modelling charge transport and electro-optical characteristics of quantum dot light-emitting diodes. *npj Comput. Mater.* **7**, 122 (2021).
 25. Bae, W. K. *et al.* Multicolored light-emitting diodes based on all-quantum-dot multilayer films using layer-by-layer assembly method. *Nano Lett.* **10**, 2368–2373 (2010).
 26. Bae, W. K. *et al.* R/G/B/Natural white light thin colloidal quantum dot-based light-emitting devices. *Adv. Mater.* **26**, 6387–6393 (2014).
 27. Anikeeva, P. O., Halpert, J. E., Bawendi, M. G. & Bulović, V. Electroluminescence from a mixed red–green–blue colloidal quantum dot monolayer. *Nano Lett.* **7**, 2196–2200 (2007).
 28. Nam, T. W. *et al.* Thermodynamic-driven polychromatic quantum dot patterning for light-emitting diodes beyond eye-limiting resolution. *Nat. Commun.* **11**, 3040 (2020).
 29. Yanoff, M. & Duker, J. S. *Ophthalmology*. (Elsevier Inc., 2009).
 30. Leschhorn, G. & Young, R. *Handbook of LED and SSL Metrology*. (Instrument Systems GmbH, 2017).
 31. Davis, K. *et al.* Effects of ambient lighting displays on peripheral activity awareness. *IEEE Access* **5**, 9318–9335 (2017).
 32. Wang, Q., Xu, H., Zhang, F. & Wang, Z. Influence of color temperature on comfort and preference for LED indoor lighting. *Optik* **129**, 21–29 (2017).

33. Delay, E. R. & Richardson, M. A. Times estimation in humans: Effects of ambient illumination and sex. *Percept. Motor. Skill* **53**, 747–750 (1981).
34. Hsieh, M. Effects of illuminance distribution, color temperature and illuminance level on positive and negative moods. *J. Asian Archit. Build.* **14**, 709–716 (2015).
35. Kuijsters, A., Redi, J., de Ruyter, B. & Heynderickx, I. Lighting to make you feel better: Improving the mood of elderly people with affective ambiances. *PLOS ONE* **10**, e0132732 (2015).

REVIEWERS' COMMENTS

Reviewer #3 (Remarks to the Author):

Authors have revised the manuscript very carefully according to the referees' comments, all the concerns have been dealt with, I think it could be accepted as it.